# On scalable and efficient training of diffusion samplers

**Minkyu Kim**[1][*]    **Kiyoung Seong**[1][*]    **Dongyeop Woo**[1]    **Sungsoo Ahn**[1]    **Minsu Kim**[1,2]

[1]Korea Advanced Institute of Science and Technology (KAIST)    [2]Mila - Quebec AI Institute

## Abstract

We address the challenge of training diffusion models to sample from unnormalized energy distributions in the absence of data, the so-called *diffusion samplers*. Although these approaches have shown promise, they struggle to scale in more demanding scenarios where energy evaluations are expensive and the sampling space is high-dimensional. To address this limitation, we propose a scalable and sample-efficient framework that properly harmonizes the powerful classical sampling method and the diffusion sampler. Specifically, we utilize Monte Carlo Markov chain (MCMC) samplers with a novelty-based auxiliary energy as a Searcher to collect off-policy samples, using an auxiliary energy function to compensate for exploring modes the diffusion sampler rarely visits. These off-policy samples are then combined with on-policy data to train the diffusion sampler, thereby expanding its coverage of the energy landscape. Furthermore, we identify primacy bias, i.e., the preference of samplers for early experience during training, as the main cause of mode collapse during training, and introduce a periodic re-initialization trick to resolve this issue. Our method significantly improves sample efficiency on standard benchmarks for diffusion samplers and also excels at higher-dimensional problems and real-world molecular conformer generation.

## 1 Introduction

Inference in unnormalized densities is a central challenge in machine learning, underlying probabilistic deep learning [19, 25] and many scientific applications [33, 9]. Traditionally, Markov chain Monte Carlo (MCMC) methods have been used, most prominently Metropolis-adjusted Langevin algorithms (MALA) [37] and Hamiltonian Monte Carlo (HMC) [15], but they incur repeated energy-gradient evaluations per sample. Amortized inference instead trains deep generative models to map noise to samples, enabling evaluation-free generation at test time and promising orders-of-magnitude speedups once the model is trained.

Researchers have recently focused on diffusion samplers, which parameterize continuous-time diffusion processes with neural networks, an approach inspired by successes in high-dimensional settings like image and text generation. The leading methods include flow-annealed importance sampling bootstrap (FAB) [30], generative flow networks (GFlowNets) [3], denoising diffusion samplers (DDS) [43], controlled Monte Carlo diffusion (CMCD) [44], and iterative denoising energy matching (iDEM) [1]. Because samples from the target distribution are unavailable, these samplers iterate between: (1) sample from the neural diffusion model, (2) query the energy, and (3) update the model to better match the target distribution.

Despite their promise, diffusion-based samplers struggle in high dimensions. Early in training, the neural proposal is effectively random and is not aligned with the energy landscape, leading to sample-inefficient exploration. This is in contrast with the classic training-free samplers, e.g., MALA, which leverage gradient information to steer proposals toward low-energy modes from the start.

---

[*]Equal contribution, correspondence to: {minkyu-kim, kiyoung.seong}@kaist.ac.kr

Techniques to improve the sample efficiency of diffusion samplers, like replay buffers [3] and local energy-guided refinements [21], yield only marginal gains and fail to overcome the poor quality of the initial diffusion samples. Indeed, He et al. [18] recently showed that nearly all effective neural samplers rely on Langevin parametrization, i.e., incorporating energy gradients at inference, which erodes the primary efficiency benefit of amortized sampling.

Moreover, diffusion samplers are prone to mode collapse: training on their own outputs leads to overfitting to dominant modes, and the model "locks in" prematurely. Reinforcement learning exploration bonuses [36] can broaden coverage, but at the cost of biasing the sampler's target distribution. Local perturbations [21] help, but require many expensive iterations in large state spaces.

**Contribution.** We propose search-guided diffusion samplers (*SGDS*), a simple yet powerful framework that enables scalable and unbiased training of diffusion samplers in high-dimensional problems. A training-free Markov-chain "Searcher" explores the target density augmented with an explicit exploration reward to discover underexplored modes. The diffusion "Learner" then distills these trajectories through the trajectory balance objective [28], preserving theoretical guarantees while incorporating exploration.

At a high level, our *SGDS* operates in two stages. **Stage 1**: the Searcher collects informative samples from the target (optionally with exploration incentives) to overcome the random initialization of the Learner. The Learner is trained off-policy via trajectory balance on a mixture of Searcher- and self-generated trajectories, rapidly improving sample efficiency. **Stage 2**: the Searcher employs random network distillation (RND) bonuses [10] to probe modes the Learner has not yet covered; the Learner then ingests these enriched trajectories using trajectory balance with weight re-initialization to counter primacy bias [32].

We show that *SGDS*, despite its simplicity, produces substantial gains over baseline diffusion samplers across benchmarks: classical Gaussian mixtures and the Manywell task; particle simulation problems like LJ-13 and LJ-55; and real-world molecules, Alanine Di-, Tri-, and Tetra-peptide. Our method significantly improves sample efficiency and scalability, marking a practical path towards high-dimensional diffusion-based inference.

## 2 Preliminaries

### 2.1 Diffusion samplers as controlled neural SDEs

Let $\mathcal{E}\colon \mathbb{R}^d \to \mathbb{R}$ be an energy function defining an unnormalized density $R(x) = \exp(-\mathcal{E}(x))$. Sampling from the corresponding Boltzmann distribution $p_{\text{target}}(x) = R(x)/Z$, with partition function $Z = \int R(x)\,dx$, can be formulated as controlling the stochastic differential equation (SDE)

$$dx_t = u_\theta(x_t, t)\,dt + g(x_t, t)\,dw_t, \qquad x_0 \sim \mu_0,\ t \in [0, 1], \tag{1}$$

where $w_t$ is standard $d$-dimensional Brownian motion, $u_\theta$ is the drift function parameterized by $\theta$ e.g., neural networks, and $g$ is the diffusion function. The goal is to choose $\theta$ such that the terminal distribution $p_1^\theta$ induced by Equation (1) matches the target, i.e., $p_1^\theta(x) \propto R(x)$.

**Euler–Maruyama discretization.** With $T$ uniform steps of size $\Delta t := 1/T$, the SDE Equation (1) is discretized via the Euler–Maruyama scheme

$$x_{t+\Delta t} = x_t + u_\theta(x_t, t)\,\Delta t + g(x_t, t)\sqrt{\Delta t}\,z_t, \qquad z_t \sim \mathcal{N}(0, I_d), \tag{2}$$

which defines Gaussian forward kernels $P_F(x_{t+\Delta t} \mid x_t; \theta)$. Analogously, one defines reference backward kernels $P_B(x_{t-\Delta t} \mid x_t)$. Common choices for $P_B$ include Brownian motion $dx_t = \beta(t)\,d\bar{w}_t$ for variance-exploding (VE) processes, the time-reversed Ornstein–Uhlenbeck (OU) kernel $dx_t = -\beta(t)x_t\,dt + \sqrt{2\beta(t)}\,d\bar{w}_t$ for variance-preserving (VP) processes, and the Brownian bridge $dx_t = \frac{x_t}{t}dt + \sigma\,d\bar{w}_t$, where $\bar{w}_t$ is time-reversed Brownian motion.

The forward and backward policies for the complete trajectory $\tau = (x_0 \to x_{\Delta t} \to \cdots \to x_1)$, denoted by $P_F(\tau; \theta)$ and $P_B(\tau \mid x_1)$, repectively, are defined as compositions of these kernels across discrete time steps:

$$P_F(\tau; \theta) = \prod_{i=0}^{T-1} P_F\big(x_{(i+1)\Delta t} \mid x_{i\Delta t}; \theta\big), \quad P_B(\tau \mid x_1) = \prod_{i=0}^{T-1} P_B\big(x_{(i-1)\Delta t} \mid x_{i\Delta t}\big). \tag{3}$$

---

**Algorithm 1** Training search-guided diffusion samplers (*SGDS*)

---

1: $Q_{\text{buffer}} \leftarrow \emptyset$; **fix** random target net $f_{\text{rnd}}$; initialize predictor $\hat{f}_\phi$ and Learner $(P_F(\tau; \theta), \log Z_\theta)$
2: **for** $r = 1, \ldots, N_{\text{round}}$ **do**                                           ▷ *outer rounds*
3:      *// Searcher: gradient-guided MCMC*
4:      $\tilde{\mathcal{E}}(x) \leftarrow \begin{cases} \mathcal{E}(x), & r = 1, \\ \mathcal{E}(x) - \alpha \left\| f_{\text{rnd}}(x) - \hat{f}_\phi(x) \right\|_2^2, & r > 1 \end{cases}$
5:      Obtain $\{x_1^{(i)}\}_{i=1}^{M_{\text{chain}}}$ and $\log \hat{Z}$ by running $M_{\text{chain}}$ parallel MCMC for $M_{\text{iter}}$ steps on $\tilde{\mathcal{E}}(x)$
6:      $Q_{\text{buffer}} \leftarrow Q_{\text{buffer}} \cup \{x_1^{(i)}, \mathcal{E}(x_1^{(i)})\}_{i=1}^{M_{\text{chain}}}$
7:      *// Learner: I inner iterations (even iterations: on-policy, odd iterations: off-policy)*
8:      **for** $i = 1, \ldots, I$ **do**
9:          **if** $i \bmod 2 = 0$ **then**                                           ▷ *on-policy*
10:             Sample $\{\tau_k\}_{k=1}^B \sim P_F(\tau; \theta)$
11:             $\mathcal{X} \leftarrow \{x_1 \text{ from } \tau_k\}$
12:         **else**                                                                  ▷ *off-policy*
13:             Sample $\mathcal{X} = \{x_1\}_{\ell=1}^{B_{\text{off}}} \sim P(\cdot \mid Q_{\text{buffer}})$
14:             Generate $\{\tau_\ell\} \sim P_B(\tau \mid x_1)$
15:         **end if**
16:         $\mathcal{L}_{\text{TB}} = \frac{1}{B} \sum_k \left[ \log \frac{Z_\theta P_F(\tau_k; \theta)}{R(x_1) P_B(\tau_k \mid x_1)} \right]^2$
17:         $\theta \leftarrow \text{Minimize}(\mathcal{L}_{\text{TB}})$                ▷ *diffusion sampler update*
18:         $\phi \leftarrow \text{Minimize}\left( \frac{1}{|\mathcal{X}|} \sum_{x_1 \in \mathcal{X}} \| f_{\text{rnd}}(x_1) - \hat{f}_\phi(x_1) \|_2^2 \right)$      ▷ *RND predictor update*
19:      **end for**
20:      Re-initialize $P_F(\cdot \mid \theta)$ but retain $\log Z_\theta$          ▷ *Periodic partial re-initialization*
21: **end for**

---

**Stochastic control of neural SDEs.** Diffusion models typically minimize the forward Kullback–Leibler (KL) divergence

$$D_{\text{KL}}\big(P_B(\tau \mid x_1)\, p_{\text{target}}(x_1) \,\|\, P_F(\tau; \theta)\, \mu_0(x_0)\big),$$

which presupposes abundant samples from $x_1 \sim p_{\text{target}}$. When such data are unavailable, e.g., in scientific domains, one may instead minimize the reverse KL divergence

$$D_{\text{KL}}\big(P_F(\tau; \theta)\, \mu_0(x_0) \,\|\, P_B(\tau \mid x_1)\, p_{\text{target}}(x_1)\big),$$

using samples from $x_1 \sim P_F$. Notable methods that optimize this objective include the path-integral sampler (PIS) [47], which employs a VE Brownian-motion reference process, and denoising diffusion samplers (DDS) [43], which use a VP OU reference process.

## 2.2   Continuous GFlowNet objective for diffusion samplers

Following Sendera et al. [39], Euler–Maruyama samplers can be interpreted as continuous generative flow networks (GFlowNets) [24]. GFlowNets [3, 4] are off-policy reinforcement-learning algorithms for sequential decision making samplers. Treating the initial state $x_0$ as a point mass at the origin, the forward policy $P_F$ acts as an agent that sequentially constructs a trajectory $\tau$. The trajectory balance (TB) criterion [28] guarantees that the density induced by $P_F$ matches the target distribution:

$$Z_\theta\, P_F(\tau; \theta) = R(x_1)\, P_B(\tau \mid x_1), \qquad \forall \tau, \tag{4}$$

where $Z_\theta$ is a learnable scalar that approximates the unknown partition function $Z$. Existing GFlowNet-based samplers [46, 39] often adopt Brownian-bridge kernels for $P_B$.

Applying the TB condition to sub-trajectory of $\tau$ yields the *sub-trajectory balance* objective [27, 35, 46]. While this variant can improve credit assignment, it estimates marginal densities at intermediate states with higher bias compared to the global TB estimates [39].

**Off-policy property of GFlowNet-based diffusion samplers.** In contrast to KL-based objectives such as PIS or DDS, using on-policy training, GFlowNet objectives can be optimized with *off-policy* trajectories drawn from any proposal distribution with full support. This flexibility enables richer exploration strategies—noisy roll-outs [24], replay buffers, and MCMC-based local search [39]—that are crucial for efficient sampling from multimodal distributions.

## 3 Method

### 3.1 Search-guided diffusion samplers (*SGDS*): overall framework

In this section, we describe the overall framework of the search-guided diffusion samplers (*SGDS*). Our *SGDS* combines the strengths of *off-policy* training from GFlowNet diffusion samplers with the exploratory power of gradient-guided MCMC. We follow the setting of Sendera et al. [39] for modeling GFlowNet-based diffusion samplers. Each *round* alternates between two roles:

**Searcher (gradient-informed MCMC).** The Searcher uses gradient information $\nabla \log \pi(x)$ to efficiently generate representative samples from the target distribution. These samples populate a replay buffer and simultaneously provide an estimate of the log partition function, $\log Z$. Exploration is guided by an intrinsic reward from random network distillation (RND) [10], which identifies underexplored modes using a form of self-supervised learning.

**Learner (diffusion sampler).** Learner, a neural diffusion sampler, is trained by minimizing trajectory balance loss [24], blending (i) *on-policy* trajectories generated from its current policy and (ii) *off-policy* trajectories replayed from the buffer. Periodic re-initialization of the Learner mitigates primacy bias, enhancing sample efficiency.

This round repeats until the Learner alone generates high-quality samples. For simple targets, training may converge within a single round, while complex targets typically benefit from multiple rounds.

The *SGDS* tackles two critical challenges in existing diffusion sampling approaches:

**Scalability.** In high-dimensional spaces, diffusion samplers frequently miss low-energy modes, as their generated samples rarely visit unexplored modes. The Searcher, operating as parallel gradient-informed chains, rapidly identifies these modes. Although the samples collected from the Searcher are biased, the trajectory balance objective enables unbiased training of the Learner.

**Sample efficiency.** Each expensive gradient evaluation is amortized across multiple Learner updates through off-policy replay. The RND-driven intrinsic rewards direct the Searcher towards under-explored areas, maximizing the informativeness of new samples. Periodic Learner re-initialization prevents overfitting to initial samples and maintains replay buffer diversity. Collectively, these components significantly enhance the efficiency of gradient computations.

Algorithmic details for each component follow in subsequent sections and Algorithm 1.

### 3.2 Searcher

The Searcher identifies low-energy modes using parallel gradient-guided Markov chains. Methods such as annealed importance sampling (AIS) [31], Metropolis-adjusted Langevin algorithms (MALA) [37], or molecular dynamics (MD) are suitable candidates. These methods generate samples by transporting prior samples in the direction of the target density (or its tempered density) via several Markov chains. We use AIS and MALA for synthetic energy functions, and MD for all-atom systems.

In the initial step of the algorithm, we run $M_{\text{chain}}$ parallel chains, estimating $\log \hat{Z}$ which is explained in Appendix A. The Searcher then stores the collected samples in a replay buffer and passes the estimated $\log \hat{Z}$ to the Learner model. In subsequent rounds, we incorporate exploration uncertainty from the Learner via intrinsic rewards for exploration, modifying the Searcher's energy landscape as:

$$\tilde{\mathcal{E}}(x) = \mathcal{E}(x) - \alpha \log r_{\text{intrinsic}}(x). \tag{5}$$

Here, $r_{\text{intrinsic}}(x)$ highlights underexplored modes based on previous Learner experiences, and the gradient is used in the drift function of SDEs. Adding a repulsive term for exploration resembles the core idea of metadynamics, which biases sampling away from the modes that have already been well captured.

**Random network distillation (RND).** To efficiently guide exploration, we employ RND [10] to quantify state novelty, steering the Searcher towards underexplored consists of a fixed, randomly initialized network $f(x)$ and a trainable predictor network $\hat{f}(x; \phi)$ trained by minimizing:

$$\mathcal{L}_{\text{RND}}(x) = \|f(x) - \hat{f}(x; \phi)\|_2^2, \tag{6}$$

and, for the Searcher in the next round, we utilize this loss as the intrinsic reward given by:

$$r_{\text{intrinsic}}(x) = \exp(\|f(x) - \hat{f}(x; \phi)\|_2^2). \tag{7}$$

High prediction errors indicate novel states. RND training uses replay buffer samples and online trajectories, assigning high novelty to underexplored modes.

### 3.3 Learner

With the replay buffer initialized by Searcher's samples, the Learner minimizes the trajectory balance objective through iterative training, combining online and replay trajectories. The training incorporates:

$$\mathcal{L}_{\text{off-policy}}(\theta) = \mathbb{E}_{\tau \sim P_B(\tau|x_1), x_1 \sim P(x_1|\mathcal{D}_{\text{buffer}})} \frac{1}{2} \left[ \log \frac{Z_\theta P_F(\tau; \theta)}{R(x_1) P_B(\tau \mid x_1)} \right]^2, \tag{8}$$

$$\mathcal{L}_{\text{on-policy}}(\theta) = \mathbb{E}_{\tau \sim P_F(\tau)} \frac{1}{2} \left[ \log \frac{Z_\theta P_F(\tau; \theta)}{R(x_1) P_B(\tau \mid x_1)} \right]^2. \tag{9}$$

Here $P(x_1 \mid \mathcal{D}_{\text{buffer}})$ denotes a *rank-based* sampling distribution [42] that assigns higher probability to lower energy samples stored in the buffer, focusing replay on promising modes.

We leverage both on-policy and off-policy training signals from online trajectories and replayed samples, with a replay ratio $\gamma$ determining the frequency of replay updates (default: $\gamma = 1$).

**Re-initialization.** Learner re-initialization mitigates primacy bias commonly observed in reinforcement learning scenarios. Primacy bias [32] refers to the model's tendency to rely excessively on early experiences, being trapped in low-reward or biased samples generated at initial stages, thereby hindering the discovery of high-reward samples and underexplored modes. Periodically re-initializing the Learner model $P_F(\cdot|\theta)$ alleviates this bias by resetting parameters strongly influenced by early samples, allowing faster adaptation to recent, higher-quality experiences. Crucially, we retain the previously learned $\log Z_\theta$ parameter and the replay buffer, preserving the accumulated knowledge while allowing the network to recalibrate based on updated experiences.

## 4  Related works

**Classical samplers.** Classical sampling approaches primarily rely on MCMC methods. This includes gradient-based algorithms like MALA [37] and HMC [15]. Annealing-based techniques, such as AIS [31] and SMC [12], introduce intermediate distributions to gradually approximate complex targets, mitigating mode collapse. While these MCMC-based methods enable sampling from the complex unnormalized density, they require long trajectories and extensive energy evaluations.

**Neural amortized inference.** Neural amortized inference methods aim to bypass costly MCMC by training neural samplers that generate approximate samples in one or a few forward passes. Diffusion-based neural samplers learn stochastic differential equations parameterized by neural networks to map simple priors to complex targets [47, 43], and GFlowNets train stochastic policies whose marginal visitation probabilities match an unnormalized density [3, 13]. Boltzmann Generators (BG) is another line of works to amortize inference, such as molecular dynamics simulation. BG utilizes normalizing flows trained on simulated data to sample from the Boltzmann distribution and estimate density, enabling statistical reweighting for unbiased estimates [34, 14, 30, 23, 41].

**Diffusion-based neural samplers.** Diffusion-based samplers aim to sample from unnormalized target distributions in data-free settings. Several approaches [47, 43, 44, 2, 5] formulate the sampling objective via KL divergence in path measure space. Akhound-Sadegh et al. [1] further introduces off-policy training via replay buffers. Recent works [8, 11] also explore controllable dynamics, offering improved exploration in complex energy landscapes. While these methods often improve mode coverage by learning reverse-time dynamics, they remain computationally intensive, hindering scalability in high-dimensional settings.

**Generative Flow Networks.** GFlowNets was originally introduced by Bengio et al. [3] and Bengio et al. [4] on discrete spaces where the probability of each outcome is proportional to a given reward signal. Subsequent extensions have connected GFlowNets to continuous space [24], enabling sampling from unnormalized densities in high-dimensional spaces [13, 29]. Recent work has also explored enhancements to off-policy training strategies [39] and incorporated local search mechanisms [21], allowing GFlowNets to more effectively navigate continuous energy landscapes. Additionally, adaptive reward design has emerged as a promising direction for improving mode coverage during training [22], especially in tasks that require structured exploration or sparse supervision.

Table 1: ELBO, EUBO, their gap, and energy calls across high-dimensional Manywell distributions. We use MALA as the local search algorithm. We consume 6M energy calls per searcher (12M total for 2 rounds) and 8M energy calls for the learner. **Bold** indicates the best performance per metric, and * indicates large absolute values of metrics.

| Method | Manywell ($d = 64$) | | | | Manywell ($d = 128$) | | | |
|---|---|---|---|---|---|---|---|---|
| | ELBO ↑ | EUBO ↓ | EUBO − ELBO ↓ | Energy calls | ELBO ↑ | EUBO ↓ | EUBO − ELBO ↓ | Energy calls |
| PIS+LP | 300.57 ± 0.37 | 347.48 ± 0.26 | 46.91 ± 0.55 | 130M | 601.01 ± 0.94 | 697.32 ± 0.49 | 96.31 ± 0.71 | 130M |
| TB+LP | 306.47 ± 0.23 | 351.98 ± 0.46 | 45.52 ± 0.51 | 180M | 612.45 ± 0.65 | 706.73 ± 2.59 | 94.28 ± 3.00 | 300M |
| FL-SubTB+LP | 306.14 ± 0.71 | 352.22 ± 0.62 | 46.08 ± 0.26 | 330M | 609.85 ± 0.48 | 709.96 ± 2.10 | 99.61 ± 1.83 | 330M |
| TB+LS+LP | 312.66 ± 2.66 | 339.34 ± 1.02 | 26.68 ± 3.37 | 320M | 592.52 ± 2.25 | 693.65 ± 1.40 | 101.81 ± 3.62 | 320M |
| TB+Expl+LP | 306.54 ± 0.23 | 351.91 ± 0.53 | 45.37 ± 0.66 | 180M | 611.98 ± 0.34 | 705.35 ± 1.05 | 93.37 ± 1.22 | 240M |
| TB+Expl+LS+LP | 300.10 ± 1.05 | 344.85 ± 0.41 | 44.75 ± 1.39 | 320M | 591.47 ± 0.36 | 694.93 ± 0.54 | 103.45 ± 0.88 | 320M |
| PIS | **321.87 ± 0.05** | 2026.11 ± 408.98 | 1704.91 ± 408.49 | 100M | **643.30 ± 0.09** | 1159.60 ± 48.53 | 516.30 ± 49.67 | 100M |
| TB | 317.35 ± 6.01 | 853.94 ± 43.35 | 544.36 ± 29.85 | 100M | 637.01 ± 2.14 | 1423.35 ± 292.15 | 786.35 ± 290.46 | 100M |
| TB+LS | 314.94 ± 4.60 | 357.40 ± 4.36 | 42.91 ± 9.15 | 290M | 573.13 ± 73.49 | 738.07 ± 10.77 | 164.95 ± 62.71 | 290M |
| TB+Expl+LS | 265.99 ± 95.39 | 361.00 ± 16.58 | 41.46 ± 15.47 | 290M | 589.49 ± 7.25 | 698.24 ± 2.81 | 108.74 ± 10.07 | 290M |
| GAFN [36] | 320.88 ± 0.36 | 573.68 ± 29.02 | 252.80 ± 30.87 | 100M | * | * | * | 100M |
| AT [22] + LP | 281.56 ± 2.21 | 353.64 ± 3.48 | 72.48 ± 2.97 | 370M | 462.61 ± 6.67 | 739.93 ± 4.97 | 277.32 ± 2.46 | 370M |
| iDEM [1] | 268.99 ± 1.22 | 414.18 ± 1.06 | 145.20 ± 1.60 | 300M | 494.28 ± 2.94 | 817.32 ± 3.22 | 323.04 ± 5.69 | 300M |
| *SGDS* | 320.25 ± 0.13 | **336.51 ± 0.11** | **16.26 ± 0.22** | **20M** | 614.41 ± 3.44 | **684.76 ± 1.30** | **70.35 ± 4.31** | **20M** |

| (a) Ground Truth | (b) *SGDS* | (c) PIS | (d) SubTB+LP | (e) TB+Expl+LS | (f) iDEM |

Figure 1: Mode coverage comparison using 2D projections of 10,000 samples on Manywell-128.

**Connection to previous works.** Using gradient-guided MCMC for improving exploration in off-policy diffusion samplers is not new. Lemos et al. [26] employed gradient-guided MCMC to populate replay buffers for GFlowNet diffusion sampler training. Sendera et al. [39] applied parallel MALA initialized from diffusion sampler states, similar to discrete local search GFlowNet methods [21]. Our approach extends the multiple-round algorithm of Lemos et al. [26], incorporating RL techniques to boost efficiency. It can be viewed as a deeper but shorter-cycle alternative to Sendera et al. [39], whose frequent diffusion-based re-initializations overly depend on sampler performance (see comparison with TB + LS at Table 1, Table 2, and Figure 4a).

Leveraging Learner uncertainty to guide exploration aligns with active learning and related GFlowNet approaches [36, 22]. Following generative augmented flow network (GAFN) [36], direct injection of intrinsic reward was effective, similar to our idea (see comparison with GAFN at Table 1). While Kim et al. [22] introduced additional neural samplers called adaptive teachers (AT) as Searchers to covers high loss region it is highly unstable in large scale due to Searcher's adversarial behavior with non-stationary objective, where our method efficiently employs MCMC-based exploration without additional neural network (see comparison with AT + LP at Table 1).

## 5 Experiments

In this section,[1] our primary goal is to demonstrate the performance and efficiency of our proposed framework through several experiments. Specifically, we aim to showcase the sample efficiency and scalability of our method, as well as to validate the effectiveness of the various training strategies we introduced. We focus on presenting results on high-dimensional tasks. In all the experiments, we use four different random seeds and average the results of each run. We provide details of experimental settings in Appendix A.4, Table 4, and Table 5, and additional results in Appendix B.

### 5.1 Main results

**Settings.** In this work, we compare the performance of our proposed framework against baselines on multiple benchmark tasks, including 40GMM, Manywell-32/64/128, LJ-13, and LJ-55. We

---

[1]Source code: https://github.com/minkyu1022/SGDS

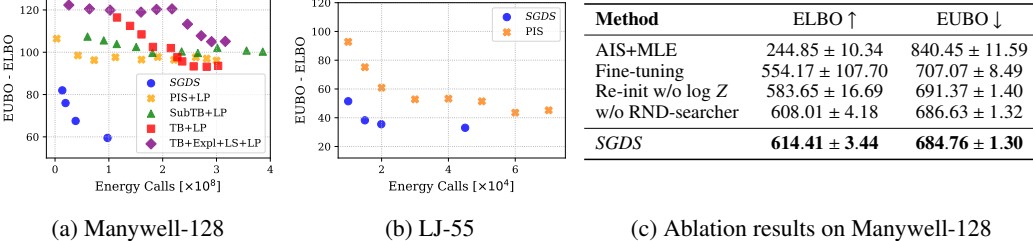

(a) Manywell-128                (b) LJ-55                (c) Ablation results on Manywell-128

Figure 2: Trade-off between EUBO–ELBO gap and energy calls in Manywell-128 (left) and LJ-55 (middle). The results of ablation study on Manywell-128 (right) show the performance of AIS using the same total energy calls with MLE amortizing, taking 2 rounds with fine-tuning instead of re-initialization, and using the Searcher with no RND rewards. All methods use 20M energy calls.

Table 2: The gap between ELBO and $\widehat{\text{EUBO}}$, (equivariant) position Wasserstein-2, energy histogram Wasserstein-2, and energy calls across Lennard-Jones potential. We denote $\widehat{\text{EUBO}}$ as the EUBO metrics calculated by the reference samples provided by [1], which are not exact samples from the target distribution. For iDEM, we reproduce the results with the hyperparameter setting which can be found in Appendix A.4. The results of Adjoint sampling are obtained from [17]. **Bold** indicates the best performance, * indicates large absolute values of metrics, - indicates inaccessible value from their papers, and "Div." indicates divergent training.

| Method | LJ-13 ($d = 39$) | | | | LJ-55 ($d = 165$) | | | |
| | $\widehat{\text{EUBO}}$ - ELBO ↓ | $\mathcal{W}_2$ ↓ | $E(\cdot)\mathcal{W}_2$ ↓ | Energy calls | $\widehat{\text{EUBO}}$ - ELBO ↓ | $\mathcal{W}_2$ ↓ | $E(\cdot)\mathcal{W}_2$ ↓ | Energy calls |
| --- | --- | --- | --- | --- | --- | --- | --- | --- |
| PIS | $2.04 \pm 0.32$ | $1.57 \pm 0.03$ | $2.10 \pm 0.46$ | 370K | $45.53 \pm 5.58$ | $4.28 \pm 0.05$ | $40.63 \pm 13.68$ | 45K |
| TB | $12.53 \pm 3.43$ | $1.71 \pm 0.07$ | $10.99 \pm 3.14$ | 370K | Div. | Div. | Div. | 45K |
| TB+Expl+LS | $11.99 \pm 0.52$ | $2.02 \pm 0.04$ | $18.69 \pm 4.49$ | 3M | * | $12.46 \pm 0.19$ | $123.02 \pm 21.72$ | 1M |
| iDEM [1] | $112.37 \pm 7.63$ | $4.27 \pm 0.02$ | $39.92 \pm 3.24$ | 300M | * | $17.17 \pm 0.32$ | $210.87 \pm 6.26$ | 120M |
| Adjoint Sampling [17] | - | $1.67 \pm 0.01$ | $2.40 \pm 1.25$ | 1M | - | $4.50 \pm 0.05$ | $58.04 \pm 20.98$ | 1M |
| *SGDS* | $\mathbf{1.53 \pm 0.25}$ | $\mathbf{1.56 \pm 0.04}$ | $\mathbf{0.89 \pm 0.19}$ | 370K | $\mathbf{33.01 \pm 0.93}$ | $\mathbf{4.25 \pm 0.08}$ | $\mathbf{32.17 \pm 12.90}$ | 45K |

evaluate methods using three metrics: the Evidence Lower Bound (ELBO), Evidence Upper Bound (EUBO) [7], and the EUBO − ELBO gap. A smaller gap between ELBO and EUBO indicates a more accurate approximation of the target distribution.

For fair comparison on the number of energy calls, we train the methods until convergence of ELBO and EUBO. To determine convergence, we evaluate based on the moving average of the metrics over the 10 consecutive evaluations, where we evaluate the model every 100 training steps. If a method does not converge within the maximum number of epochs, we report the metrics at the final step.

**Baselines.** For the manywell potential, the baselines are primarily selected based on their strong performance demonstrated in prior work [39], as well as their methodological relevance [36, 22] or their different framework [1]. Specifically, iDEM [1] utilizes trajectories of length $T = 1,000$ for SDE integration, whereas other baselines, including PIS [47], TB [28], AT [22], and GAFN [36], employ shorter diffusion trajectories ($T = 100$) with distinct optimization objectives. We further evaluate enhanced variants of these methods incorporating LP, such as PIS+LP, TB+LP, and FL-SubTB+LP, along with exploration-enhanced (+Expl) or local search (+LS) variants introduced by Sendera et al. [39]. We describe the details of the abbreviations related to baselines in Appendix A.1. For the LJ potentials, we compare against Adjoint Sampling [17], as well as iDEM, PIS, TB, and its variant TB+Expl+LS. We omit LP-based methods due to their divergent training.

**Results.** As shown in Table 1 and Table 2, our proposed framework consistently achieves superior performance across all high-dimensional tasks: Manywell-64, Manywell-128, and LJ-55. Especially, our method demonstrates the best trade-off between performance and efficiency of energy call.

In Figure 1, one can observe that our method better captures the modes in Manywell-128 when compared to the baselines. As illustrated in Figure 2a and Figure 2b, even increasing the energy budget of baselines does not allow them to surpass the performance of our proposed approach. Also, as shown in Figure 3, our framework generates high quality samples with low energy. Furthermore, for the LJ-55 potential, the distribution of interatomic distances is similar to the ground truth

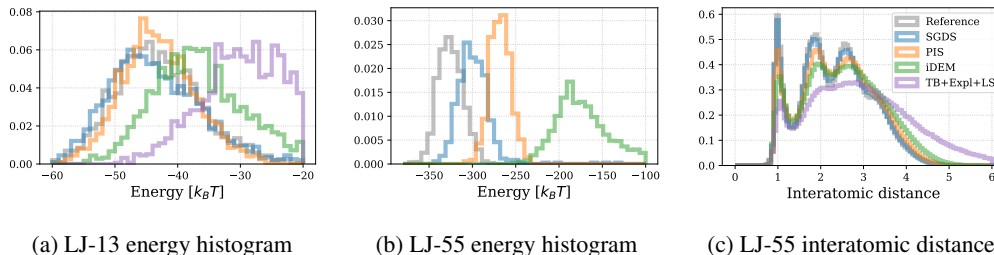

(a) LJ-13 energy histogram      (b) LJ-55 energy histogram      (c) LJ-55 interatomic distance

Figure 3: Histograms for LJ-13/55 energy densities and LJ-55 interatomic distances.

distribution. Additionally, our method obtains competitive results with significantly fewer samples in lower-dimensional tasks such as 40GMM, Manywell-32 (see Appendix B.1), and LJ-13 (see Table 2).

## 5.2 Ablation study

**MCMC sampler with the same budget.** In our method, we consume energy calls during both Searcher sampling and Learner training. To evaluate the efficiency of the Learner's on/off-policy mixing training scheme, we conduct a controlled comparison where the total energy call budget (20M) is entirely allocated to AIS on Manywell-128. We run 200 chains on the trajectories with $T = 10,000$ for AIS. As a result, even though high-reward samples were collected using AIS with much longer trajectories, the MLE Learner failed to perform amortized learning as shown in Figure 2c.

**Periodic re-initialization and pre-trained flow.** We perform an ablation study to evaluate two design components of our method when proceeding to the next training round: (1) re-initializing the Learner model, and (2) retaining the pre-trained $\log Z$ parameter from the previous round. Specifically, to assess the benefit of re-initialization in mitigating primacy bias, we compare our method against a fine-tuning baseline where the second-round Learner continues training from the first-round model weights without re-initialization. To isolate the effect of retaining the estimated $\log Z$ value, we compare against a variant where the $\log Z$ parameter is also re-initialized at the start of the second round. As shown in Figure 2c, our full method outperforms both ablation variants, confirming that re-initialization is beneficial for mitigating primacy bias, and that employing the $\log Z$ parameter across rounds leads to better training stability and performance.

**Novelty-based reward in Searcher sampling.** We assess the effectiveness of incorporating the novelty-based intrinsic reward derived by RND [10] into the Searcher sampling process in later training rounds. In our framework, starting from the second round, the Searcher sampler drives prior samples in the direction of the target distribution and exploration signal derived from a previously trained RND module, which prioritizes underexplored modes by the Learner sampler. These dynamics guide the Searcher to focus sampling efforts on modes that remain novel and close to the target distribution across rounds. As shown in Figure 2c, at the end of round 2, our RND-augmented approach yields a smaller EUBO–ELBO gap compared to a way of repeating the same Searcher sampling without exploration. These results demonstrate that using intrinsic rewards to adaptively bias Searcher sampling toward novel modes improves overall distributional coverage across rounds.

## 5.3 Application to molecular conformer generation.

We also consider three real-world systems, Alanine Di-, Tri-, and Tetra-peptide, consisting of 23, 33, 43 atoms in vacuum at a temperature of 300K. While some previous works show promising results in sampling conformation of Alanine Dipeptide, they rely on low-dimensional descriptors such as rotatable torsion angles [45]. Solving these three peptides at all-atom resolutions remains a challenge for existing diffusion-based neural samplers.

**Settings.** To accurately evaluate molecular energies, we employ TorchANI [16], a PyTorch implementation of ANI deep learning potentials trained on quantum-mechanical reference data. For the Searcher, we run four parallel 55ps Langevin dynamics simulations under the TorchANI potential. In the first round, simulations are performed at 600 K to efficiently sample slow degrees of freedom; in the second round, we use 300 K to capture faster motions and collect high-reward samples. The Learner and RND models use the $E(3)$-equivariant graph neural network (EGNN) architecture [38] based on atomic coordinates. We provide details in Appendix A.4.

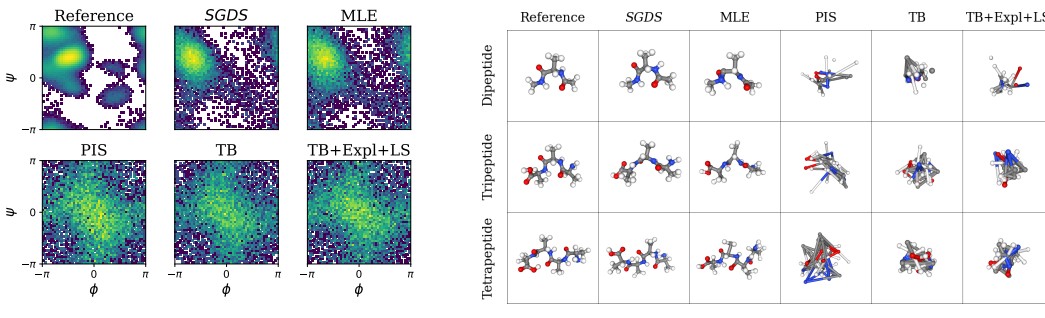

|  (a) Ramachandran plots | (b) 3D visualization of sampled conformations |

Figure 4: Qualitative results of methods in three peptides. (a) Ramachandran plot of Alanine Dipeptide with two backbone torsion angles $(\phi, \psi)$ and (b) 3D visualization of generated conformations.

**Baselines.** We compare our method with maximum-likelihood estimation (MLE) as well as diffusion-based neural samplers: PIS, TB, and TB+Expl+LS. We generate reference ensemble using 100 ns Langevin-dynamics simulation at 300K. In MLE, the likelihood of forward path distribution is maximized over the samples from backward path distribution. To match the number of energy calls of MLE with our method, we collect 2.5 times more samples than our method, using the same searcher as our method without RND. TB+Expl+LS utilizes the same langevin dynamics for local search algorithm. We omit the LP methods since they have large absolute values of EUBO and ELBO. We exclude comparison with Volokhova et al. [45], as they consider only rotatable torsion angles, and with Midgley et al. [30], which employs a discrete normalizing flow on internal coordinates, whereas our method utilizes a diffusion model in atomic coordinates.

**Results.** As shown in Table 3, our method outperforms diffusion-based neural samplers and MLE on three peptides. As illustrated in Figure 4, both our method and MLE capture the free energy landscape and generate physically plausible conformations by leveraging high-fidelity samples from the Langevin dynamics Searcher. By contrast, PIS, TB, and TB+Expl+LS fail to reconstruct the target free-energy surface or to produce realistic structures, since their forward policies insufficiently explore the complex landscape. We note that the Langevin dynamics used

Table 3: EUBO-ELBO gaps on peptides in units of $10^3$. **Bold** indicates the best performance. * indicates large absolute values of metrics, and "Div." indicates divergent training.

| Method | Dipeptide | Tripeptide | Tetrapeptide |
|---|---|---|---|
| PIS (1M) | $84.65 \pm 9.93$ | $35.23 \pm 2.43$ | * |
| TB (1M) | Div. | Div. | Div. |
| TB+Expl+LS (3M) | $18.86 \pm 0.19$ | $6.41 \pm 0.34$ | Div. |
| MLE (1M) | $17.41 \pm 0.14$ | $6.11 \pm 0.02$ | $29.07 \pm 0.56$ |
| *SGDS* (1M) | $\mathbf{17.11 \pm 0.16}$ | $\mathbf{5.60 \pm 0.03}$ | $\mathbf{26.60 \pm 0.21}$ |

in the Searcher yields higher-quality samples than those obtained by local searches of TB+Expl+LS from forward-policy outputs. Furthermore, our approach refines the biased samples from the high-temperature Searcher through an unbiased TB objective, improving performance compared to MLE.

## 6   Conclusion

We have proposed a scalable and sample-efficient sampling framework *SGDS* that integrates an MCMC Searcher with a diffusion Learner. By leveraging high-quality samples from replay buffers and training the Learner model via on/off-policy TB objectives, our method effectively bridges classical sampling with neural amortization. The inclusion of novelty-based intrinsic rewards by RND further enhances the exploration of the Searcher, enabling informed guidance to underexplored modes throughout multiple rounds.

Our work opened promising directions for integrating learning-based amortization with classical sampling, particularly for tasks where both diversity and precision are crucial. Future extensions include designing multi-agent search systems that leverage classical sampling methods for cooperative strategic exploration in high-dimensional spaces and developing advanced off-policy learning schemes, such as adaptive filtering strategies for the replay buffer.

## Acknowledgements

This work was partly supported by Institute for Information & communications Technology Planning & Evaluation(IITP) grant funded by the Korea government(MSIT) (RS-2019-II190075, Artificial Intelligence Graduate School Support Program(KAIST)), National Research Foundation of Korea(NRF) grant funded by the Ministry of Science and ICT(MSIT) (No. RS-2022-NR072184), GRDC(Global Research Development Center) Cooperative Hub Program through the National Research Foundation of Korea(NRF) grant funded by the Ministry of Science and ICT(MSIT) (No. RS-2024-00436165), and the Institute of Information & Communications Technology Planning & Evaluation(IITP) grant funded by the Korea government(MSIT) (RS-2025-02304967, AI Star Fellowship(KAIST)).
Minsu Kim was supported by KAIST Jang Yeong Sil Fellowship.

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

# A  Experiment details

Code is available at .

And the reference samples can be downloaded from .

## A.1  Description of notations

PIS (Path Integral Sampler) [47], TB (Trajectory Balanced) [28], and FL-SubTB (Forward-Looking SubTB) [39] denote the types of objectives. GAFN (Generative Augmented Flow Networks) [36] is a GFlowNet-based method that directly injects intrinsic rewards into TB loss. AT (Adaptive Teacher) [22] introduces an additional neural sampler (the Teacher) trained to focus on high-loss regions of the Student. LP (Langevin Parametrization) [39] refers to a drift construction technique where the model's drift term is combined with the gradient of the target energy function, helping the trained dynamics to better follow the target energy landscape. LS (Local Search) [39] denotes a refinement step where a short Markov chain guided by the target energy gradient is applied from the final state of the generated trajectory to improve sample quality. Expl (Exploration) [39] represents a noise-scheduling technique applied during trajectory generation, in which additional stochasticity is injected more in the early training phase to promote broad exploration and gradually reduced over time for exploitation later.

## A.2  MCMC samplers for Searcher

**Annealed importance sampling (AIS).** Annealed importance sampling (AIS) [31] is an MCMC sampling method for estimating the partition functions of target distributions. AIS bridges between an easy-to-sample initial distribution $\pi_0(x)$ and a target distribution $\pi_T(x)$ through a sequence of intermediate distributions $\{\pi_t(x)\}_{t=0}^T$, where $T$ is the length of a trajectory or chain. Each intermediate distribution $\pi_t(x)$ typically has the form:

$$\pi_t(x) \propto \pi_0(x)^{1-\beta_t} \pi_T(x)^{\beta_t}, \quad 0 = \beta_0 < \beta_1 < \cdots < \beta_T = 1, \tag{10}$$

where $\{\beta_t\}$ is a predefined annealing schedule, and we use $\beta_t = \frac{t}{T}$ in our framework. AIS generates samples through an MCMC transition kernel at each intermediate distribution with the following SDE simulation:

$$\mathrm{d}x_t = \nabla \log \pi_t(x_t)\mathrm{d}t + \sqrt{2}\mathrm{d}W_t, \tag{11}$$

where $\nabla \log \pi_t(x_t) = (1 - \beta_t)\nabla \log \pi_0(x_t) + \beta_t \nabla \log \pi_T(x_t)$ is the score function of the annealed distribution (unnormalized). Then it accumulates importance weights given by:

$$w_{\mathrm{AIS}} = \prod_{t=1}^{T} \frac{\pi_t(x_{t-1})}{\pi_{t-1}(x_{t-1})}, \tag{12}$$

and the expectation of these weights provides an unbiased estimator of the partition function ratio between $\pi_T(x)$ and $\pi_0(x)$:

$$\frac{Z_T}{Z_0} \approx \frac{1}{N} \sum_{i=1}^{N} w^{(i)}, \tag{13}$$

where $w^{(i)}$ is the importance weight computed for the $i$-th AIS trajectory, and $N$ is the total number of trajectories. We compute the unbiased estimation of the log scale of the partition function for Manywell experiments by

$$\log \hat{Z}_T = \log \frac{1}{N} \sum_{i=1}^{N} w^{(i)}, \tag{14}$$

where $\log Z_0 = 0$ because the initial distribution is Gaussian in our framework.

**Metropolis-Adjusted Langevin Algorithm (MALA).** The Metropolis-Adjusted Langevin Algorithm (MALA) is an MCMC method that uses the gradient of the energy function to generate samples from

a target distribution $\pi(x)$. MALA starts by sampling initial states $x_0 \sim \pi_0(x_0)$, where $\pi_0(\cdot)$ is some proposed initial distribution (in most cases, $\mathcal{N}(0, \sigma^2 I)$). It then iteratively proceeds transition from $x_t$ to $x_{t+1}$ by simulating the following Langevin dynamics:

$$\mathrm{d}x_t = -\nabla\mathcal{E}(x_t)\mathrm{d}t + \sqrt{2}\mathrm{d}W_t, \qquad (15)$$

Here, $x_t$ is the current state at time $t$, $W_t$ denotes the standard Brownian motion, and $\mathcal{E}(x)$ is the energy function of target distribution $\pi(x)$, i.e. $-\nabla\mathcal{E}(x_t) = \nabla \log \pi(x_t)$.

The proposed sample $x_{t+1}$ is then accepted or rejected according to the Metropolis-Hastings acceptance probability:

$$\alpha = \min\left\{1, \frac{\pi(x_{t+1})q(x_t \mid x_{t+1})}{\pi(x_t)q(x_{t+1} \mid x_t)}\right\}, \qquad (16)$$

where $q(\cdot \mid \cdot)$ denotes the Gaussian transition density induced by the Langevin proposal:

$$x_{t+1} = x_t - \nabla\mathcal{E}(x_t)\Delta t + \sqrt{2\Delta t} \cdot z, \quad z \sim \mathcal{N}(0, I). \qquad (17)$$

The step size $\Delta t$ is a key factor influencing the quality of sampling. For all tasks, we utilize the scheduling of step size, by comparison between the current acceptance rate and the target acceptance rate (57.4%). We use MALA as Searcher on 40GMM, LJ-13, and LJ-55.

Also, since a MALA trajectory forms a Markov chain, consecutive samples are still correlated and therefore $\{x_i\}_{i=1}^N$ are not strictly i.i.d. To reduce the most severe correlations we discard the first $M_{\text{burn-in}}$ iterations as burn-in and use all subsequent states directly. We then compute a rough estimate

$$\log \hat{Z} = \log\left[\frac{1}{N} \sum_{i=1}^N \exp\left(-\mathcal{E}(x_i)\right)\right], \qquad (18)$$

where this estimator is biased since $x_i \sim \pi$ ideally and $\mathbb{E}_\pi\left[\exp\left(-\mathcal{E}(x)\right)\right] = Z \int \pi^2(x)dx < Z$. Despite the bias, the estimation can provide a numerically reasonable heuristic value for the initialization of the Learner's $\log Z_\theta$.

**Underdamped Langevin dynamics.** For MCMC Searchers of three peptides, we adopt underdamped Langevin dynamics as our molecular dynamics (MD). This framework combines deterministic forces with stochastic fluctuations, which is essential for accurately capturing thermal motion and inertial effects of the molecules. The resulting dynamics are governed by the following system of stochastic differential equations:

$$\begin{aligned} \mathrm{d}x_t &= v_t \, \mathrm{d}t, \\ \mathrm{d}v_t &= -M^{-1}\nabla\mathcal{E}(x_t) \, \mathrm{d}t - \gamma v_t \, \mathrm{d}t + \sqrt{2\gamma k_B T M^{-1}} \, \mathrm{d}W_t. \end{aligned} \qquad (19)$$

Here, $x_t$ is the position at time $t$, $v_t$ is the velocity, $M$ is the mass matrix (symmetric positive definite), $\mathcal{E}(x)$ is the potential energy function, and $\nabla\mathcal{E}(x_t)$ is its gradient with respect to position, i.e., the negative force. The parameter $\gamma$ is the friction coefficient, $k_B$ is the Boltzmann constant, $T$ is the absolute temperature, and $W_t$ denotes standard Brownian motion.

For peptides, we use underdamped Langevin dynamics as MD with high temperature(600K). We use Euler-Maruyama integration to discretize the Langevin dynamics. As in MALA, we compute $\log \hat{Z}$ for the initialization of $\log Z_\theta$ in Learner, using Equation (18).

## A.3 Metrics

In this subsection, we formally define the evaluation metrics used to assess the Learner's quality. All metrics are derived from the same importance-weight formulation based on the target partition function.

We begin with the exact log partition function $\log Z$, which can be written using forward-path importance sampling. Let $\tau = (x_0, x_{\Delta t}, \ldots, x_1)$ denote a sample trajectory drawn from the forward

policy $P_F(\tau)$, and let $R(x_1)$ be the reward associated with the final state $x_1$. Then, the partition function can be expressed as

$$\log Z = \log \mathbb{E}_{\tau \sim P_F(\tau)} \left( \frac{R(x_1) P_B(\tau \mid x_1)}{P_F(\tau)} \right), \tag{20}$$

where $P_B(\tau \mid x_1)$ is the backward policy conditioned on the final state.

Since directly optimizing this quantity is intractable, we use two surrogate bounds. The first is the evidence lower Bound (ELBO), defined as

$$\text{ELBO} = \mathbb{E}_{\tau \sim P_F(\tau)} \left[ \log \frac{R(x_1) P_B(\tau \mid x_1)}{P_F(\tau)} \right]. \tag{21}$$

By Jensen's inequality, ELBO is always a lower bound on the true $\log Z$. It is commonly used as a training objective and can reflect how well the forward policy $P_F$ concentrates on high-reward trajectories. However, ELBO can be misleading in practice. A high ELBO does not necessarily imply that all important modes are captured, as the forward policy may collapse to a small subset of modes while still achieving high reward [7].

To address this limitation, we also evaluate the evidence upper Bound (EUBO), which flips the sampling distribution:

$$\text{EUBO} = \mathbb{E}_{\tau \sim P_B(\tau)} \left[ \log \frac{R(x_1) P_B(\tau \mid x_1)}{P_F(\tau)} \right]. \tag{22}$$

Unlike ELBO, EUBO acts as a diagnostic metric. It is an upper bound of $\log Z$ and penalizes missing probability mass. EUBO is driven to penalize missing probability mass and therefore exposes mode-collapse that ELBO may hide [7]. And then, true $\log Z$ is consequently bounded by two bounds, i.e., $\text{ELBO} \le \log Z \le \text{EUBO}$.

A smaller gap between the two bounds yields a tighter estimate of $\log Z$, making this gap a useful indicator of the Learner's sampling quality.

Table 4: Searcher configurations of SGDS

| Benchmark | 40GMM | Manywell 32 | Manywell 64 | Manywell 128 | LJ-13 | LJ-55 | Peptides |
|---|---|---|---|---|---|---|---|
| Type | MALA | AIS | AIS | AIS | MALA | MALA | MD |
| # of Chains | 300 | 60K | 60K | 60K | 16 | 1 | 4 |
| Chain length | 4K | 100 | 100 | 100 | 4K | 10K | 110K |
| Burn-in | 2K | - | - | - | 2K | 4K | 10K |
| init. step size | 1e-3 | 1e-3 | 1e-3 | 1e-3 | 1e-5 | 1e-5 | 0.5fs |

Table 5: Learner configurations of SGDS

| Benchmark | 40GMM | Manywell 32 | Manywell 64 | Manywell 128 | LJ-13 | LJ-55 | Peptides |
|---|---|---|---|---|---|---|---|
| Brownian bridge std ($\sigma$) | 10.0 | 1.0 | 1.0 | 1.0 | 0.2 | 0.2 | 0.2 |
| Buffer size | 600k | 60k | 60k | 60k | 50K | 10K | 800K |
| Batch size | 300 | 300 | 300 | 300 | 32 | 4 | 16 |
| Architecture | MLP | MLP | MLP | MLP | EGNN | EGNN | EGNN |
| hidden dim | 256 | 256 | 256 | 256 | 64 | 64 | 128 |
| # of layers | 2 | 2 | 2 | 2 | 5 | 5 | 5 |
| RND weight | 100 | 100 | 100 | 100 | 10 | 1 | 10 |

## A.4 Experimental setup

We reproduce iDEM [1] with modified hyperparameter settings to ensure a comparable number of energy calls. Specifically, we adjust the number of MC samples for score estimation and the number of iterative loops. Furthermore, we exclude the additional refinement steps originally applied to the LJ-55 potential in iDEM to maintain consistency across all evaluated methods.

For the diffusion-based neural samplers on synthetic benchmarks, we follow the setup of [39].

**Gaussian mixture model with 40 modes (40GMM).** Training proceeds in one or two rounds. Our framework achieves competitive performance against baselines even with only a single round, and

shows marginal improvement with a second round. We use MALA as the Searcher, running 300 parallel chains of length 4K, discarding the first 2K steps as burn-in. We maintain a target acceptance rate of 57.4% through step size scheduling, resulting in a total of 2.4M energy evaluations. We use the Gaussian prior with a standard deviation of 21.0 for MALA.

All methods adopt the PIS architecture [47, 39], with a joint network consisting of a two-layer MLP with 256 hidden dimensions. The RND network consists of three layers in the predictor network and the target network, with 256 hidden dimensions. We adopt Brownian bridges as the backward process, with a Brownian motion coefficient of 10.0. We run 25K epochs in both the first round and the second round.

**Manywell distributions.** We proceed with one or two rounds for training on Manywell distributions. We use AIS as the Searcher, running 60K parallel chains (3K chains * 20 iterations) of length 100, only taking the final step samples. We use the Gaussian prior with a standard deviation of 1.0.

All methods adopt the PIS architecture [47, 39], with a joint network consisting of a two-layer MLP with 256 hidden dimensions. The RND network consists of three layers in the predictor network and the target network, with 256 hidden dimensions. We adopt Brownian bridges as the backward process, with a Brownian motion coefficient of 1.0. We run 25K epochs in the first round and 30K in the second round.

**Lennard-Jones (LJ) potentials.** Training proceeds in two rounds. We use MALA as the Searcher for two rounds: in LJ-13 we run 16 parallel chains of length 4K corresponding to 64K energy evaluations, discarding the first 2K steps as burn-in and retaining 57.4% accepted samples among remaining 32K samples; in LJ-55 we run a single chain of length 10K corresponding to 10K energy evaluations, discarding the first 4K steps and retaining 57.4% accepted samples among remaining 6K samples. We use the Gaussian prior with a standard deviation of 1.75 for MALA.

All methods utilize five EGNN layers with 64 hidden dimensions. Following [20, 23], we design an E(3)-equivariant generative model initialized from a Dirac delta at the origin, using a mean-free forward transition kernel in inference. The RND network comprises three layers in the predictor network and two in the target network. We adopt Brownian bridges as the backward process for diffusion-based neural samplers, with a Brownian motion coefficient of 0.2. For LJ-13, we run 5K epochs in the first round and 10K in the second round; for LJ-55, 10K and then 20K epochs.

Specifically, we note that the reported performance of the iDEM on Table 2 differs from the original paper [1] due to adjustments, except $\sigma_{max}$ and $\sigma_{min}$ of the noise scheduling, made to avoid significant discrepancies in energy call usage compared to our method. We reduce the EGNN hidden dimension to 64 and the batch size to 8, and limit the total number of training epochs, including both inner and outer loops, to 15K accordingly. And while the latest iDEM codebase employs 10 steps of Langevin dynamics refinement before evaluation, particularly for LJ-55, we omit this step for fair comparison and instead set the number of samples for MC estimation to 1K. While iDEM reports a lower bound of $\log Z$ computed via importance sampling with its learned proposal density $q(x)$ given by OT-CFM model, we omit this result in our tables. We compute the lower bound based on trajectory-level estimators without training auxiliary models, i.e., CFM. Thus, our reported values are not directly comparable to those from iDEM.

Additionally, in LJ-55, we maximize the log-likelihood of the forward path distribution under the backward process for the first 5K epochs of each round, discretizing backward paths from Brownian bridges initialized with empirical samples collected by Searchers. We also use randomized time scheduling introduced in [6] for our method. We train PIS at a learning rate of $1e - 4$, TB at a learning rate of $2e - 4$, and SGDS at a learning rate of $5e - 4$. We use 4 and 32 batch sizes for all methods except PIS in LJ-13 and LJ-55, respectively. For PIS, we halve these sizes due to the memory limitation required by the forward SDE computational graph.

**Peptides.** We perform two rounds of search using under-damped Langevin dynamics. In each round, we run four parallel simulations of 55 ps each, with a time step of 0.5 fs, requiring 440K energy evaluations. We discard the first 5 ps of each trajectory as burn-in, then collect 400K samples. Each simulation starts from the same initial position drawn from a Dirac delta distribution, with all initial velocities set to zero. We integrate equations of motion using the Euler–Maruyama integrator, set the friction coefficient $\gamma = 1$, and use temperature $T = 600K$ for the first round Searcher and $T = 300K$ for the second round Searcher.

Similar to LJ potentials, all models utilize five EGNN layers with 128 hidden dimensions. We use a Dirac delta prior distribution at the origin and a mean-free forward transition kernel to guarantee $E(3)$-equivariance of the marginal density in inference. The Learner network comprises five EGNN layers, while the predictor network and target network in the RND framework contain three and two layers, respectively. As in LJ potentials, we use the Brownian motion coefficient of 0.2. We run 10K epochs in the first round and 20K epochs in the second. As in LJ-55, we maximize the log-likelihood for the first 5K epochs each round. We also utilize randomized time scheduling for our method. We train PIS at a learning rate of $1e-4$ and all other methods at $5e-4$. We use a 16 batch size for all methods except PIS, which uses an 8 batch size due to the memory limitation required by the forward SDE computational graph.

In inference time, we follow [23]. We first align the topology of generated samples with the target bond graph since the architecture and machine learning potential have a degree of freedom in atom ordering. We first match the bond graphs of generated samples with a given bond graph of interest and then correct the chirality of the generated sample to fit the target molecular configuration. The generated sample is rejected if the bond graph is not isomorphic to the target bond graph.

## A.5 Task details

**40-Component Gaussian Mixture Model (40GMM).** The 40-component Gaussian Mixture Model (GMM) consists of a mixture distribution of 40 Gaussian components, each characterized by a distinct mean vector $\mu_i$. The energy function for the GMM is defined as:

$$\mathcal{E}(x) = -\log\left(\frac{1}{n}\sum_{i=1}^{n}\mathcal{N}(x; \mu_i, \sigma^2 I)\right),$$

where $n = 40$, the weight of each $i$-th Gaussian component is the same, and $\mathcal{N}(x; \mu_i, \sigma^2 I)$ is the probability density function of the multivariate Gaussian distribution.

**ManyWell distributions.** The Manywell potential describes a high-dimensional energy landscape containing multiple wells (local minima), each representing stable states with distinct energy levels. The energy function of Manywell distribution is given by:

$$\mathcal{E}(x) = \sum_{k=1}^{n}(x_{2k-1}^4 - 6x_{2k-1}^2 - \frac{1}{2}x_{2k-1} + \frac{1}{2}x_{2k}^2) + C,$$

where $n = d/2$ is the number of wells, and $d$ is the dimensionality of the landscape. Adjusting the dimensionality $d = 2n$ allows varying the number of wells and complexity, creating tasks like Manywell-32, Manywell-64, and Manywell-128.

**Lennard-Jones (LJ) potentials.** The Lennard-Jones potential models the interactions between particles. The energy function is defined as:

$$\mathcal{E}(x) = 2\kappa \sum_{1 \le i < j \le N} \epsilon\left[\left(\frac{\sigma}{r_{ij}}\right)^{12} - 2\left(\frac{\sigma}{r_{ij}}\right)^{6}\right] + \frac{\lambda}{2}\sum_{i=1}^{N}\|r_i - r_{cm}\|^2, \tag{23}$$

where $\epsilon$ and $\kappa$ are parameters defining the depth of the potential well and the energy factor, respectively. $r_{ij} = \|x_i - x_j\|$ represents the Euclidean distance between particles $i$ and $j$. $\sigma$ is the characteristic distance at which the potential between two particles vanishes, often interpreted as the van der Waals radius. In our experiments, we set all parameters to 1.0, i.e., $\kappa = \epsilon = \sigma = \lambda = 1.0$. Adjusting the number of particles creates tasks such as LJ-13 and LJ-55, increasing the complexity of the particle interactions and resulting in a rugged energy landscape.

**TorchANI potential for peptides.** We leverage TorchANI [16], a PyTorch implementation of ANI deep-learning potentials trained on quantum-mechanical reference data, to accurately calculate molecular energies. It provides transferable machine learning potential trained on organic molecules for efficient energy and force evaluation with accuracy comparable to density-functional theory (DFT). In particular, TorchANI excels at modeling small peptides.

Table 6: ELBO, EUBO, their gap, and Energy calls on 40GMM and Manywell-32.

| Method | 40GMM ($d = 2$) | | | | Manywell ($d = 32$) | | | |
|---|---|---|---|---|---|---|---|---|
| | ELBO ↑ | EUBO ↓ | Gap ↓ | Energy calls | ELBO ↑ | EUBO ↓ | Gap ↓ | Energy calls |
| PIS+LP | $-1.32 \pm 0.07$ | $2.42 \pm 0.20$ | $3.75 \pm 0.22$ | 300M | $160.83 \pm 0.41$ | $180.49 \pm 4.76$ | $19.66 \pm 4.78$ | 300M |
| TB+LP | $-0.35 \pm 0.03$ | $0.53 \pm 0.04$ | $0.87 \pm 0.03$ | 160M | $161.42 \pm 0.40$ | $195.89 \pm 8.14$ | $34.37 \pm 8.15$ | 300M |
| FL-SubTB+LP | $-0.36 \pm 0.01$ | $0.58 \pm 0.08$ | $0.94 \pm 0.07$ | 260M | $160.74 \pm 0.15$ | $215.93 \pm 4.52$ | $55.19 \pm 4.52$ | 330M |
| TB+LS+LP | $-0.38 \pm 0.03$ | $0.32 \pm 0.02$ | $0.69 \pm 0.02$ | 320M | $162.95 \pm 0.08$ | $166.30 \pm 0.11$ | $3.35 \pm 0.14$ | 320M |
| TB+Expl+LP | $-0.37 \pm 0.01$ | $0.32 \pm 0.02$ | $0.69 \pm 0.02$ | 300M | $160.76 \pm 0.13$ | $215.92 \pm 14.90$ | $55.16 \pm 14.90$ | 300M |
| TB+Expl+LS+LP | $-0.37 \pm 0.01$ | $0.34 \pm 0.02$ | $0.71 \pm 0.02$ | 320M | $162.97 \pm 0.06$ | $166.25 \pm 0.10$ | $3.28 \pm 0.12$ | 320M |
| PIS | $-2.03 \pm 0.22$ | $55.48 \pm 10.71$ | $57.50 \pm 9.02$ | 100M | $159.71 \pm 1.70$ | $333.79 \pm 3.98$ | $174.08 \pm 4.33$ | 100M |
| TB | $-1.35 \pm 0.04$ | $99.04 \pm 6.01$ | $100.40 \pm 5.67$ | 100M | $160.58 \pm 0.87$ | $439.28 \pm 166.52$ | $278.70 \pm 166.49$ | 100M |
| TB+LS | $-0.38 \pm 0.03$ | $0.83 \pm 0.46$ | $1.21 \pm 0.38$ | 290M | $163.12 \pm 0.10$ | $166.05 \pm 0.12$ | $2.93 \pm 0.16$ | 290M |
| TB+Expl+LS | $-0.38 \pm 0.05$ | $0.58 \pm 0.34$ | $0.96 \pm 0.34$ | 290M | $160.87 \pm 3.31$ | $168.27 \pm 1.49$ | $7.40 \pm 3.63$ | 290M |
| GAFN | * | * | * | N/A | $161.02 \pm 0.05$ | $282.40 \pm 2.02$ | $121.38 \pm 2.02$ | 100M |
| iDEM | $-2.14 \pm 0.45$ | $12.75 \pm 3.67$ | $14.89 \pm 3.70$ | 300M | $142.23 \pm 0.40$ | $211.56 \pm 2.53$ | $69.33 \pm 2.56$ | 300M |
| *SGDS* (round 1) | $\mathbf{-0.40 \pm 0.01}$ | $\mathbf{0.33 \pm 0.02}$ | $\mathbf{0.73 \pm 0.02}$ | **6M** | $162.49 \pm 0.05$ | $166.60 \pm 0.01$ | $4.11 \pm 0.05$ | **9M** |
| *SGDS* (round 2) | $\mathbf{-0.40 \pm 0.03}$ | $\mathbf{0.33 \pm 0.05}$ | $\mathbf{0.73 \pm 0.05}$ | **12M** | $\mathbf{162.63 \pm 0.01}$ | $\mathbf{166.48 \pm 0.03}$ | $\mathbf{3.85 \pm 0.03}$ | **20M** |

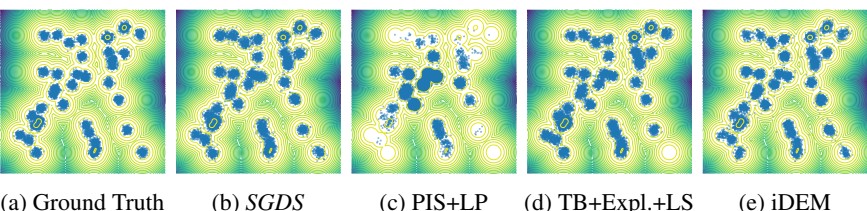

   (a) Ground Truth   (b) *SGDS*   (c) PIS+LP   (d) TB+Expl.+LS   (e) iDEM

Figure 5: Mode coverage comparison on 40GMM.

## B   Additional experimental results

### B.1   Low-dimensional standard benchmarks

**Baselines and settings.** We benchmark our framework on two standard low-dimensional tasks: 40GMM and Manywell-32. Consistent with the high-dimensional experiments, we report ELBO, EUBO, their gap, and the number of energy calls required during training. We employ the same baseline methods and trajectory configurations (including trajectory length and training objectives) as in the high-dimensional settings. We provide detailed configurations, including diffusion scales for each task, in Table 5.

**Results.** As demonstrated in Table 6, our method achieves competitive performance on lower-dimensional standard tasks, producing EUBO and ELBO metrics comparable to the strongest baselines, while using significantly fewer energy calls. On the 40GMM task, despite some baselines reporting strong ELBO and EUBO scores, they notably fail to capture the mode located at the bottom-right corner (see Figure 5). In contrast, our framework reliably identifies all modes without sacrificing performance metrics. We report both the first-round and second-round performances of our method, showing that our method attains robust performance on low-dimensional tasks even in the first round, with a slight but consistent improvement observed in the second round.

### B.2   Debiasing of Learner from MCMC Searcher

To address potential biases inherent in MCMC sampling due to finite-length chains, our framework incorporates both off-policy TB training using samples from the Searcher and on-policy TB training. This design choice aims to mitigate biases arising from the Searcher samples alone by enabling the Learner model to adjust toward the target distribution.

To evaluate whether the Learner effectively debiases the samples collected by the Searcher, we compare kernel density estimations (KDE) of samples obtained by the AIS Searcher with those generated by the on/off-policy TB Learner on Manywell distributions. Figure 6 illustrates these KDE comparisons across dimensions 32, 64, and 128.

Due to the varying mode masses assigned in Manywell distributions, even when AIS successfully covers all modes with limited budgets, it struggles to precisely capture the relative mode masses.

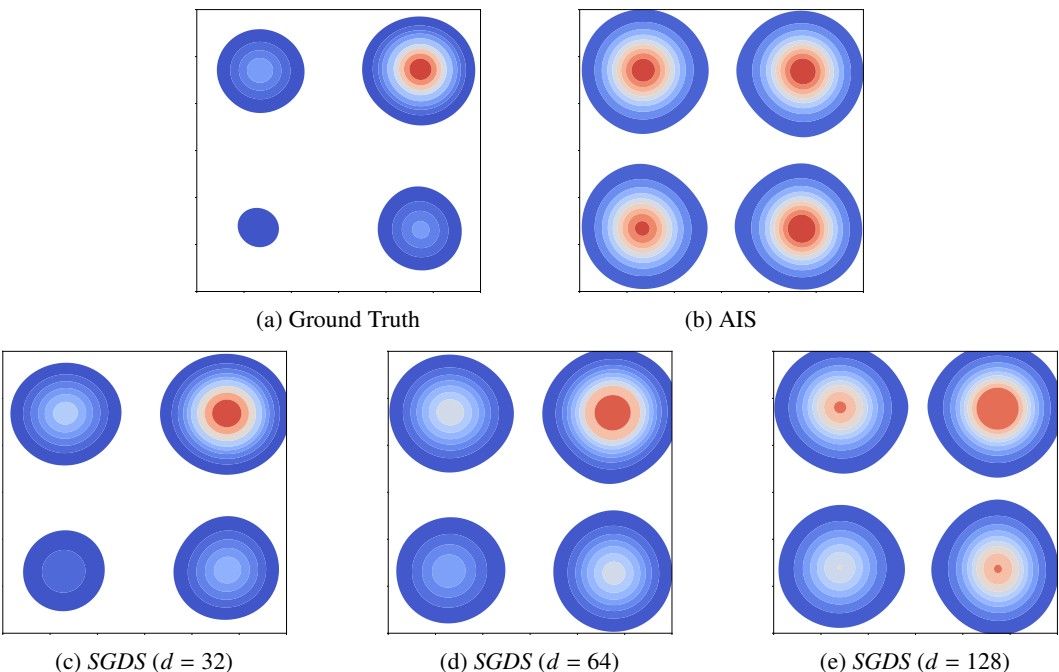

| (a) Ground Truth | (b) AIS |
|---|---|

| (c) *SGDS* ($d = 32$) | (d) *SGDS* ($d = 64$) | (e) *SGDS* ($d = 128$) |
|---|---|---|

Figure 6: KDE figures of AIS($T = 100$), ours, and true samples on Manywell-32/64/128.

Table 7: EUBO–ELBO gap at 20% of the second-round training for different RND weight values.

| RND weight | $10^0$ | $10^1$ | $10^2$ | $10^3$ |
|---|---|---|---|---|
| Manywell-128 | 549.54 | 542.60 | **538**.**79** | 570.49 |
| LJ-13 | 3.12 | **2**.**67** | 3.90 | 4.77 |
| LJ-55 | **34**.**76** | 36.72 | 44.78 | 51.03 |
| ALDP | 17,531.95 | **17**, **381**.**04** | 17,417.75 | 17,410.98 |

In contrast, the KDE of the samples generated by the Learner aligns more closely with the true density, effectively reflecting the relative importance of different modes. This result highlights the effectiveness of combining on- and off-policy training to achieve better density approximation than relying solely on finite-budget AIS samples.

### B.3 RND Weight Calibration

To evaluate the sensitivity of our method to the choice of the RND weight, we conducted additional experiments across approximately four different RND weight values per task. The results, summarized in Table Table 7, show that the performance is largely robust to this hyperparameter. The calibration of the RND weight is straightforward: we run 20% of the second-round training to estimate performance and select a reasonable value. As shown in the ELBO–EUBO gap measured at this stage, reported for Manywell-128, LJ potentials, and peptides, the variation across different RND weights is minor, indicating that extensive hyperparameter tuning is unnecessary.

Overall, these results suggest that the method performs consistently across a wide range of RND weights, with negligible degradation in stability or performance. Although a more systematic tuning strategy could be explored in future work, the current approach provides reliable results with minimal calibration effort.

### B.4 Consistency with parallel tempering MD

We further validate the consistency of our method by replacing the Searcher with parallel tempering MD [40]. Parallel tempering (or replica exchange) MD is an enhanced sampling method that runs

multiple independent simulations (or replicas) at different temperatures and periodically attempts to exchange them, allowing low-temperature simulations to overcome energy barriers and escape local minima. Also, we align the number of energy evaluations by collecting additional data for MLE, ensuring a fair comparison. Table 8 shows the ELBO, EUBO, and their gap, demonstrating that our method achieves better metrics than MLE. This result indicates that our approach consistently works even when combined with advanced sampling techniques such as parallel tempering.

Table 8: Comparison of main metrics between parallel tempering + MLE (forward KL) training and our method with parallel tempering Searcher. We align the number of energy calls by collecting more data for MLE.

| Method | ELBO $[\times 10^3]\uparrow$ | EUBO $[\times 10^3]\downarrow$ | Gap $[\times 10^3]\downarrow$ |
|--------|------|------|------|
| MLE | $520.71 \pm 0.02$ | $538.03 \pm 0.00$ | $17.32 \pm 0.02$ |
| *SGDS* | $521.03 \pm 0.01$ | $538.02 \pm 0.00$ | $16.99 \pm 0.01$ |

## C   Limitations

While our framework demonstrates strong empirical performance, several limitations remain.

First, the effectiveness of intrinsic rewards from RND depends on careful tuning of the novelty scale parameter $\alpha$. Poorly calibrated $\alpha$ can overly emphasize exploration, producing noisy or irrelevant samples, or conversely yield overly conservative exploration. This could be mitigated by employing adaptive strategies that dynamically adjust $\alpha$ during sampling based on diversity metrics or exploration progress signals.

Additionally, the quality of samples provided by the Searcher sets a fundamental exploration limit. If the Searcher fails to adequately explore challenging modes, the Learner will inevitably inherit these limitations, particularly in high-barrier energy landscapes. Introducing enhanced exploration strategies, such as parallel tempering or more advanced proposal schemes like HMC, could improve coverage of hard-to-sample modes.

