# OpenReview forum: "On scalable and efficient training of diffusion samplers"
_NeurIPS.cc/2025/Conference — NeurIPS 2025 poster_

### Official Review · Reviewer_HM7Y · 2025-06-24

**Clarity:** 3
**Significance:** 2
**Originality:** 2
**Rating:** 4
**Confidence:** 4

**Summary:**

The paper proposes a scalable sample-efficient method for generating samples from an unnormalized density with increased exploration capabilities. To authors propose to collect off-policy samples using MCMC using the target energy with an exploration bonus that is computed using random network distillation. This data is then used to learn a diffusion model with off-policy TB loss and combined with on-policy data collected by the diffusion model. Additionally, the authors suggest to periodically reset the weights of the networks to prevent primacy bias. The method is evaluated on several toy and many-body problems.

**Questions:**

- What does LP stand for?
- What is the benefit of this intrinsic motivation term, compared to just running MCMC on a higher target temperature? The latter also should encourage exploration.
- Did the authors try the Log-variance loss instead of TB?

**Ethical Concerns:**

["NO or VERY MINOR ethics concerns only"]

**Final Justification:**

I remain somewhat skeptical about the novelty and practical impact of the work, but I appreciate the authors' efforts and will increase my rating to 4.

**Limitations:**

yes

**Quality:**

3

**Strengths And Weaknesses:**

Strenghts

- The paper tackles the very important problem of improving exploration in diffusion-based sampling.
- The paper uses mode-covering metrics such as EUBO.
- The approach seems very sample efficient.

Weaknesses

- The RND weight varies between several orders of magnitude between different tasks (Table 5), suggesting the need for an intensive hyperparameter search per task to obtain a good exploration behavior. An ablation study would help to understand the sensitivity of the method for different values of the RND weight. However, the reviewer acknowledges that this was already included as a limitation of the method.
- I currently doubt that the method outperforms parallel tempering / replica exchange MCMC / MD + forward KL training. It is well known that MD can easily obtain ground truth samples from ALDP. However, the performance of SGDS on ALDP doesn’t seem convincing to me due to significant differences in the Ramachandran histogram.
- The ablation study on ManyWell shows that the RND searcher does not significantly boost performance.
- A comparision to Adjoint sampling seems missing, which has shown to be very scalable and sample efficient as well.

[a] No Trick, No Treat: Pursuits and Challenges Towards Simulation-free Training of Neural Samplers

---

> ### Author Rebuttal · Authors · 2025-07-30
>
> Dear Reviewer HM7Y,
>
> Thank you for your thoughtful and encouraging review. We appreciate your recognition of the contributions and evaluations of our work. Below, we address your comments in detail.
>
> ### **W1: The RND weight varies between several orders of magnitude for different tasks. An ablation study would help to understand the sensitivity of the method for different RND weights.**
>
> To resolve your concern, we provide experimental results that verify that intensive hyperparameter search is not needed, and the results from different RND weights are not sensitive.
>
> Taking these factors into account, we conducted experiments across approximately four different RND weight values per task to ensure stable training and meaningful results. The calibration of the RND weight $\alpha$ is important but simple: run 20% of the second round to measure the performance.
>
> ###### **Table 1. ELBO-EUBO gap at 20% of the second round for Manywell-128, LJ potentials, and ALDP.**
>
> | RND weight | $10^{0}$ | $10^{1}$ | $10^{2}$ | $10^{3}$ |
> | - | - | - | - | - |
> | Manywell-128 | 549.54 | 542.60 | **538.79** | 570.49 |
> | LJ-13 | 3.12 | **2.67** | 3.90 | 4.77 |
> | LJ-55 | **34.76** | 36.72 | 44.78 | 51.03 |
> | ALDP | 17,531.95 | **17,381.04** | 17,417.75 | 17 410.98|
>
> We found that reasonable values could be selected with minimal tuning, though a more systematic approach could be explored in future work.
>
> ---
>
> ### **W2: I currently doubt that the method outperforms forward KL training combined with one of parallel tempering, replica exchange MCMC, or MD.**
>
> To dispel your doubt, we show that our method outperforms MLE (or forward KL) training on samples from parallel tempering (or replica exchange) MD in Table 2.
>
> ###### **Table 2. ELBO-EUBO gap comparison between parallel tempering + MLE (or forward KL) training and ours with parallel tempering searcher. We align the number of energy calls by collecting more data for MLE.**
>
> | Method | ELBO $[\times 10^{3}] \uparrow$ | EUBO $[\times 10^{3}] \downarrow$ | Gap $[\times 10^{3}]\downarrow$ |
> | - | - | - | - |
> | MLE | $520.71\pm0.02$ | $538.03\pm0.00$ | $17.32\pm0.02$ |
> | Ours | $\mathbf{521.03\pm0.01}$ | $\mathbf{538.02\pm0.00}$ | $\mathbf{16.99\pm0.01}$ |
>
> ---
>
> ### **W3: The ablation study on ManyWell shows that the RND searcher does not significantly boost performance.**
>
> We believe the RND searcher significantly boosts the performance since the reported values are in log scale. The presence of the RND searcher results in an improvement of $6.4$ of ELBO in the ablation study, which actually represents $\exp(6.4) \approx 403.43$ times better performance in lower bounding the marginal probability of the Learner generating the true sample.
>
> ### **W4: A comparison to Adjoint Sampling seems missing, which has shown to be very scalable and sample efficient as well.**
>
> We reproduced Adjoint Sampling algorithm and conducted the experiments to compare SGDS and Adjoint Sampling on LJ-13 and LJ-55 potentials. The results show that SGDS outperforms Adjoint Sampling in the Wasserstein distances of positions and energy histograms.
>
> ###### **Table 3. Comparison of Equivariant position Wasserstein-2 and energy histogram Wasserstein-2 between SGDS and Adjoint Sampling on LJ-13. The parameters of LJ-13 energy function are set to be the same as the settings in our experiments.**
>
> | LJ13 | $\mathcal{W}_2 \downarrow$ | $E(\cdot)$ $\mathcal{W}_2 \downarrow$  | # of energy calls |
> | - | - | - | - |
> | Adjoint Sampling (reproduced) | 1.95 | 10.76 | 1M |
> | Adjoint Sampling (reported) | 1.67 | 2.40 | 1M |
> | SGDS (ours) | **1.63** | **0.7589** | 370K |
>
> ###### **Table 4. Comparison of Equivariant position Wasserstein-2 and energy histogram Wasserstein-2 between SGDS and Adjoint Sampling on LJ-55.**
>
> | LJ55 | $\mathcal{W}_2 \downarrow$ | $E(\cdot)$ $\mathcal{W}_2 \downarrow$  | # of energy calls |
> | - | - | - | - |
> | Adjoint Sampling (reported) | 4.50 | 58.04 | 1M |
> | SGDS (ours) | **4.17** | **23.0422** | 45K |
>
> We reproduced Adjoint Sampling by closely following the public codes and variable settings, but the training on LJ-55 exploded. So we instead compare our method against the value reported in Adjoint Sampling paper for LJ-55 task.
>
> ---
>
> ### **Q1: What does LP stand for?**
>
> LP stands for **Langevin Parameterization**—a neural‑network design where the input includes the energy gradient as mentioned in Section 1, and the network outputs a scaled and biased version of this gradient, thus embedding a Langevin‑style inductive bias [2].
>
> ---
>
> ### **Q2: What is the benefit of this intrinsic motivation term, compared to just running MCMC on a higher target temperature?**
>
> The intrinsic-motivation term benefits from the ability to incorporate previously visited samples to better guide the exploration process.
>
> ---
>
> ### **Q3: Did the authors try the Log-variance loss instead of TB?**
>
> Yes, we did. To provide a clean and fair comparison for your question, we conducted experiments using four different seeds across 40GMM and Manywell-32/64,/128. Table 5 is the experiment results of comparing the EUBO-ELBO gap between TB and Log-variance for the Learner's objective:
>
> ###### **Table 5. Comparison of EUBO-ELBO gap between TB and Log-variance on 40GMM, Manywell-32/64/128.**
>
> | Energy | 40GMM | Manywell-32 | Manywell-64 | Manywell-128 |
> | - | - | - | - | - |
> | Log-variance | 24.79 | 74.41 | 95.95 | 392.46 |
> | TB (ours) | **0.73** | **3.85** | **16.26** | **70.35** |
>
> Here, we believe that TB outperforms Log-variance since it estimates $\log Z$ using samples from the Searcher, while Log-variance estimates $\log Z$ on-the-fly using a smaller number of samples collected from the Learner.
>
> ---
>
> [1] Sugita et al., "Replica-exchange molecular dynamics method for protein folding", Chemical physics letters 1999.
>
> [2] Sendera et al., "Improved off-policy training of diffusion samplers", NeurIPS 2024.
>
> ---
>
> We hope our responses have addressed your questions and concerns. Please let us know if there is anything we may have missed or if any points require further clarification.

---

> > ### Comment · Reviewer_HM7Y · 2025-08-04
> > **Acknowledgement**
> >
> > I thank the authors for their response. While I remain somewhat skeptical about the novelty and practical impact of the work, I appreciate the authors' efforts and will increase my rating accordingly.

---

> > > ### Author Response · Authors · 2025-08-05
> > >
> > > We sincerely thank the reviewer for the thoughtful evaluation and constructive feedback, which will be valuable for improving the clarity and impact of our work. We would include the experimental results about several RND weights and the comparison with Adjoint Sampling in the future manuscript.

---

### Official Review · Reviewer_fmfd · 2025-06-25

**Clarity:** 3
**Significance:** 3
**Originality:** 2
**Rating:** 5
**Confidence:** 3

**Summary:**

The paper proposes search-guided diffusions samplers (SGDS) to sample from unnormalized target densities.
The method trains a diffusion sampler via trajectory balance from two sources: Forward trajectories from the sampler (on-policy), and backward trajectories starting from points obtained from an MCMC searcher (off-policy).
The method is evaluated on several synthetic datasets (40GMM, Manywell), as well as atomic systems (LJ, Alanine Dipeptide) and achieves superior accuracy and mode coverage compared to prior work.

**Questions:**

# Questions

- Why is there such a large discrepancy between TB+LS and SGDS? When running TB+LS with different hyperparams (fewer local searches with more depth) would you effectively recover SGDS (less minor improvements such as weight re-init and RND)
- How are the MCMC chains for the searcher initialized? Would it make sense to draw the initial states from from forward trajectories?
- SGDS appears to be exceptionally efficient with regards to the number of energy evaluations. Why does it require so much fewer evaluations (for example compared to TB+LS)? Is it correct that SGDS only consumes energy evaluations during search (1 per chain and MCMC step) and on-policy steps (B evals per opt step), but none during off-policy steps?

# Minor points

- Eq (3): The indices of the product in $P_B$ seem to be off by one
- Equations after line 84 and 86: Shouldn't $P_F(\tau, \theta)$ also be conditioned on $x_0$ (analogous to $P_B$ being conditioned on $x_1$)? If I understand correctly you later choose $\mu_0$ to be a dirac delta at the origin but at this point (and Eq. 3) it seems unintuitive.

**Ethical Concerns:**

["NO or VERY MINOR ethics concerns only"]

**Final Justification:**

I keep my initial assessment that this is a solid paper, which outperforms existing sampling techniques.
My initial concerns were primarily around readability of the paper and qualitative comparison against existing techniques: The authors have promised to address both in the final paper.

**Limitations:**

yes

**Quality:**

3

**Strengths And Weaknesses:**

# Strengths

- Comprehensive evaluation with strong performance of SGDS across experiments
- Ablation study of proposed improvements

# Weaknesses

This appears to be a collection of various existing "tricks" (longer searches, weight re-initialization after rounds, RND-based penalization of already discovered modes) yielding overall a strong paper. I would primarily encourage to more clearly discuss the intuition behind these improvements, in particular when comparing to similar existing variants.

- The method seems to be primarily an incremental improvement on existing work, in particular Sendera et al (which the authors cite and compare against approriately). For single round training (where re-initialization and random network distillation do not apply) the only key difference appears to be that SGDS runs fewer/longer local searches vs. the frequent/short local searches of Sendera et al. It would be helpful to qualitatively discuss these differences to highly related methods more thoroughly.
- While the manuscript gives a good overview of the field, its heavy use of abbreviations makes reading key tables such as Tab. 1 hard to read for readers not intimately familiar with the field. Either naming the abbreviations in the table caption or (perhaps better) a short glossary in the supplement would be very helpful. As an example (+Expl) is only defined via a reference to Sendera et al, which in turn itself does not contain an easy-to-find definition.

---

> ### Author Rebuttal · Authors · 2025-07-30
>
> Dear Reviewer fmfd,
>
> Thank you for your thoughtful and encouraging review. We appreciate your recognition of the clarity of our experiments and evaluations. Below, we address your comments in detail.
>
> ### **W1: The method seems to be primarily an incremental improvement on existing work, in particular, Sendera et al. It would be helpful to qualitatively discuss the differences.**
>
> Our contribution is a conceptual shift in a mixture of training schemes compared to prior works. While existing diffusion samplers rely on full online learning [1,2] or use a buffer dependent on the current model's performance [3,4], SGDS utilizes both offline guidance from an independent MCMC-based searcher and online learning from the diffusion model itself. This is a meaningful improvement in the context of the training structure.
>
> ---
>
> ### **W2: It is difficult to read the tables because of their heavy use of abbreviations.**
>
> To resolve your concern, we plan to incorporate the following clarifications for Table 1 in the future manuscript as an extension of the description written in Section 5.1.
>
> * **PIS (Path Integral Sampler)** [1], **TB (Trajectory Balanced)** [5], and **FL-SubTB (Forward-Looking SubTB)** [6] denote the types of objectives. **GAFN (Generative Augmented Flow Networks)** [7] is a GFlowNet-based method that directly injects intrinsic rewards into TB loss. **AT (Adaptive Teacher)** [8] is a framework that introduces an additional neural sampler (the Teacher) trained to focus on high‑loss regions of the Student.
>
> * **LP (Langevin Parametrization)** [4] refers to a drift construction technique where the model’s drift term is combined with the gradient of the target energy function, helping the trained dynamics to better follow the target energy landscape.
>
> * **LS (Local Search)** [4] denotes a refinement step where a short Markov chain, guided by the target energy gradient, is applied from the final state of the generated trajectory to improve sample quality.
>
> * **Expl (Exploration)** [4] represents a noise-scheduling technique applied during trajectory generation, in which additional stochasticity is injected more in the early training phase to promote broad exploration and gradually reduced over time for exploitation later.
>
> ---
>
> ### **Q1-1: Why is there such a large discrepancy between TB+LS and SGDS?**
>
> The differences of RND reward, initial distribution, and search length make a large discrepancy between TB+LS and SGDS.
>
> While TB+LS relies on only Langevin dynamics with the target energy function, SGDS utilizes RND reward together for sample-efficient exploration, which is one of the key components in our framework.
>
> And TB+LS always seeds each walk from the current diffusion sample [4]; if that sampler is low‑quality in training, exploration is constrained, while SGDS does not depend on the current training states.
>
> Also, TB+LS search is too short to reach high‑reward regions on larger, sharper tasks. SGDS extends each walk, letting trajectories climb farther before termination.
>
> These three factors synergistically contribute to the substantial performance gap between SGDS and TB+LS.
>
>
> ### **Q1-2: When running TB+LS with different hyperparams, would you effectively recover SGDS (except for re-init and RND reward)?**
>
> No, they share some motivations, yet the methods are fundamentally different in where to begin the search. TB+LS starts search from the samples generated by the current Learner; if that sampler is low‑quality in training, exploration is constrained, while SGDS does not depend on the current training states.
>
> Additionally, we conducted an experiment on Manywell-128 to address your question of how TB+LS performs with a double-fewer freuquency but double deeper chains.
>
> ###### **Table 1. ELBO-EUBO gap of SGDS and TB+LS with different hyperparameters for local search on Manywell-128.**
>
> | Energy | Gap $\downarrow$ | LS cycle | Chain length | Energy calls |
> | - | - | - | - | - |
> | TB+LS | 164.95 | 100 | 200 | 290M |
> | TB+LS | 96.45 | 200 | 400 | 290M |
> | SGDS | **70.35** | - | - | 20M |
>
> Fewer and deeper local search with TB could make TB+LS better, but SGDS is still superior in performance and sample efficiency.
>
> ---
>
> ### **Q2-1: How are the MCMC chains for the searcher initialized?**
>
> We initialize the MCMC chains for the searcher with a Gaussian prior in MALA and AIS, and relaxed positions starting from a fixed position in pdb file in MD.
>
> ### **Q2-2: Would it make sense to draw the initial states from forward trajectories?**
>
> Drawing the initial states from forward trajectories does not make sense in our algorithm. We initialize the MCMC chains with a Gaussian prior. Seeding the search with forward trajectories reduces exploration and biases it toward the learner’s current distribution—especially problematic when the learner is immature. This choice helps explain why SGDS outperforms TB + LS, which draws its initial states from the diffusion sampler’s forward policy.
>
> ---
>
> ### **Q3-1: Why does SGDS require fewer evaluations (for example, compared to TB+LS)?**
>
> SGDS requires fewer energy evaluations than other methods since it consumes energy calls for collecting the Searcher samples, and it recycles the samples and rewards for off-policy training. For example, TB+LS methods waste too many samples for searching high-reward regions as they initialize the local search start point too frequently using the Learner.
>
> ### **Q3-2: Is it correct that SGDS only consumes energy evaluations during search (1 per chain and MCMC step) and on-policy steps (B evals per opt step), but none during off-policy steps?**
>
> Yes.
>
> ---
>
> ### **Minor1: Eq (3): The indices of the product in $P\_B$ seem to be off by one.**
>
> ### **Minor2 : Equations after line 84 and 86: Shouldn't $P\_F(\tau;\theta)$ also be conditioned on $x\_0$ (analogous to $P\_B$ being conditioned on $x\_1$)?**
>
> We thank you for pointing out the minor typos in the equations. We will correct $\sum_{i=0}^{T-1}P_B(x_{(i-1) \Delta t} | x_{i \Delta t})$ to $\sum\_{i=1}^{T}P_B(x\_{(i-1) \Delta t} | x\_{i \Delta t})$, and $P\_F(\tau;\theta)\mu\_0(x\_0)$ to $P\_F(\tau|x\_0;\theta)\mu\_0(x\_0)$ in the future manuscript.
>
> ---
>
> [1] Zhang et al., "Path Integral Sampler: a stochastic control approach for sampling", ICLR 2022.
>
> [2] Vargas et al., "Transport meets variational inference: Controlled Monte Carlo diffusions", ICLR 2024.
>
> [3] Akhound-Sadegh et al., "Iterated Denoising Energy Matching for Sampling from Boltzmann Densities", ICML 2024.
>
> [4] Sendera et al., "Improved off-policy training of diffusion samplers", NeurIPS 2024.
>
> [5] Malkin et al., "Trajectory balance: Improved credit assignment in gflownets", NeurIPS 2022.
>
> [6] Zhang et al., "Diffusion Generative Flow Samplers: Improving learning signals through partial trajectory optimization", ICLR 2024.
>
> [7] Pan et al., "Generative augmented flow networks", ICLR 2023.
>
> [8] Kim et al., "Adaptive teachers for amortized samplers", ICLR 2025.
>
> ---
>
> We hope our responses have addressed your questions and concerns. Please let us know if there is anything we may have missed or if any points require further clarification.

---

> > ### Comment · Reviewer_fmfd · 2025-08-05
> >
> > I thank the authors for their thorough response!
> > I would encourage them to include the qualitative comparison to other methods (in particular TB+LS) in the manuscript. If there is insufficient space in the main text, then at least in the supplement.

---

> > > ### Author Response · Authors · 2025-08-05
> > >
> > > We appreciate the reviewer’s constructive suggestion. In the future manuscript, we will incorporate the qualitative comparisons to other methods (particularly TB+LS) in the experimental sections.

---

### Official Review · Reviewer_DVcd · 2025-07-07

**Clarity:** 4
**Significance:** 3
**Originality:** 3
**Rating:** 5
**Confidence:** 3

**Summary:**

In this paper, the author's improve diffusion sampling--the application off diffusion models to sampling from a distribution only with access to its unnormalised density function--specifically targetting settings where computing the gradient of the log density is expensive and in high-dimensional settings where exploration is key. They propose SGDS which combines off-policy training with biased MCMC algorithms that searches for representative examples.

**Questions:**

- Does the paper's limited consideration of real-world datasets reflect what is standard in the literature?
- In the limitations, it is discussed that calibration of alpha is important. How simple is this callibration?

**Ethical Concerns:**

["NO or VERY MINOR ethics concerns only"]

**Final Justification:**

I do not know a great deal about developments in this area but the paper is well-written and has convinced me that they have made a contribution to an important field.

**Limitations:**

Yes, but the limitations are in the appendix. I would encourage the authors to find space for this in the main body

**Quality:**

3

**Strengths And Weaknesses:**

Strengths:
- The paper is well-written and provides a good overview of the literature and the tools that they borrow in their algorithm.
- Diffusion samplers are an important and recent class of algorithms and papers targetting practical improvements to these algorithms, like this paper, are important.
- The toy example experiments indicate a clear benefit of SGDS over existing algorithms.

Weaknesses:
- I am not overly familiar with the ALDP example but I think it would be good to compare with some papers that target this task (it seems that the comparison is mainly against generic sampling or diffusion sampling algorithms). For example, if the baseline from Midgley et al. was included, this would help.
- The contributions made in this paper appear to be the use of two existing methods to the setting of diffusion samplers. Due to my lack of knowledge of the literature, it is not immediately clear to me how much of a novel contribution this is.

---

> ### Author Rebuttal · Authors · 2025-07-30
>
> Dear Reviewer DVcd,
>
> Thank you for your thoughtful and encouraging review. We greatly appreciate your recognition of the clarity of our writing, the importance of our field, and the significance of our experimental results. Below, we address your comments in detail.
>
> ### **W1: It would be good to compare with some papers that target this task, e.g., the baseline from Midgley et al. [1].**
>
> Thank you for the suggestion. We did not include FAB [1] in our comparison because, as noted in [6], FAB cannot amortize sampling and relies on annealed importance sampling, which demands many energy evaluations. Instead, we compare against Adjoint Sampling method from [6], which also targets small peptides like Alanine Dipeptide. As shown in Table 1, when the model is trained solely on Alanine Dipeptide, our method outperforms Adjoint Sampling.
>
> ###### **Table 1. Comparison of Wasserstein-2 distance of positions and energy histogram on Alanine Dipeptide.**
> | ALDP | $\mathcal{W}_2 \downarrow$ | $E(\cdot)$ $\mathcal{W}_2 \downarrow$  |
> | - | - | - |
> | Adjoint Sampling | 33.29 | 599.74 |
> | SGDS (ours) | $\mathbf{32.90}$ | $\mathbf{487.54}$ |
>
> ---
>
> ### **W2: Unclear novelty in combining two existing methods.**
>
> We resonate with your concern, but we also believe that our method is not just a combination of two methods.
> The diffusion‑sampling field sits at the crossroads of diffusion models, probabilistic inference, stochastic control, and reinforcement learning, so progress often comes from *properly hybridizing* techniques that each community describes in its own language.
>
> * Prior work has imported existing **stochastic‑control** formulations into diffusion models [1],
> * utilized existing **replay buffer** auxiliary [6],
> * applied existing **local credit‑assignment** tricks [7] from RL, and
> * mapped existing **search algorithms** onto diffusion sampling [8].
>
> Our contribution follows this line by combining sample-efficient RL techniques with MCMC inside a diffusion sampler, which is not previously explored and therefore a meaningful addition to the literature.
>
> ---
>
> ### **Q1: Does the paper's limited consideration of real-world datasets reflect what is standard in the literature?**
>
> Yes. In fact, our work expands the standard benchmarks [7-9], i.e., GMM, Lennard-Jones, and Manywell, by additionally including Alanine Dipeptide experiments. In the future, we expect researchers to further the algorithms and apply them to real-world problems, e.g., fine-tuning biomolecular foundation models like BioEmu [10] with energy function evaluations.
>
> ---
>
> ### **Q2: In the limitations, it is discussed that calibration of alpha is important. How simple is the calibration of RND weight?**
>
> We simply choose from four possible RND weights (1, 10, 100, 1000) based on the performance at 20% of the second round, while the performance has a dependency on the RND weight.
>
> To alleviate your concern, we conducted experiments across approximately four different RND weight values per task to ensure stable training and meaningful results in Table 2.
>
> ###### **Table 2. ELBO-EUBO gap at 20% of the second round for Manywell-128, LJ potentials, and ALDP.**
>
> | RND weight | $10^{0}$ | $10^{1}$ | $10^{2}$ | $10^{3}$ |
> | - | - | - | - | - |
> | Manywell-128 | 549.54 | 542.60 | **538.79** | 570.49 |
> | LJ-13 | 3.12 | **2.67** | 3.90 | 4.77 |
> | LJ-55 | **34.76** | 36.72 | 44.78 | 51.03 |
> | ALDP | 17,531.95 | **17,381.04** | 17,417.75 | 17 410.98 |
>
> We found that reasonable values could be selected with minimal tuning, though a more systematic approach could be explored in future work.
>
> ---
>
> [1] Midgley et al., "Flow annealed importance sampling bootstrap", ICLR 2023.
>
> [2] Noe et al., "Boltzmann generators: Sampling equilibrium states of many-body systems with deep learning", Science, 2019.
>
> [3] Jing et al., "Torsional diffusion for molecular conformer generation", NeurIPS 2022.
>
> [4] Hassan et al., "Et-flow: Equivariant flow-matching for molecular conformer generation", NeurIPS 2024.
>
> [5] Havens et al., "Adjoint Sampling: Highly Scalable Diffusion Samplers via Adjoint Matching", ICML 2025.
>
> [6] Akhound-Sadegh et al., "Iterated Denoising Energy Matching for Sampling from Boltzmann Densities", ICML 2024.
>
> [7] Zhang et al., "Diffusion Generative Flow Samplers: Improving learning signals through partial trajectory optimization", ICLR 2024.
>
> [8] Sendera et al., "Improved off-policy training of diffusion samplers", NeurIPS 2024.
>
> [9] Zhang et al., "Path Integral Sampler: a stochastic control approach for sampling", ICLR 2022.
>
> [10] Lewis et al., "Scalable emulation of protein equilibrium ensembles with generative deep learning", Science 2025.
>
> ---
>
> We hope our responses have addressed your questions and concerns. Please let us know if there is anything we may have missed or if any points require further clarification.

---

> > ### Comment · Reviewer_DVcd · 2025-08-05
> >
> > ## W1
> >
> > If I understand your response correctly, you have decided to not engage in a comparison with Midgley et al. (presumably also Volokhova et al). in this experiment—and continue to refuse to compare with these works—because they are expensive to use because they "cannot amortize sampling". Unless I am not understanding your response, I do not think this is a good reason to compare with these works in some capacity. For example, is it not possible to compare with these works for a smaller scale example, to validate that your method holds up compared to these?
> >
> > ## Q2
> > Thank you for these experiments. I encourage the authors to include this somewhere in the paper/appendix
> >
> > ## Other
> > I am satisfied with the other responses and appreciate the time taken by the authors.

---

> > > ### Author Response · Authors · 2025-08-05
> > >
> > > We thank the reviewer for their valuable comment.
> > >
> > > To promptly address your additional question, we incorporate a comparison between our method and FAB, referencing the performance reported by Akhound-Sadegh et al. [6] on LJ-13 and LJ-55 potentials.
> > > We compared the Wasserstein-2 distance of positions on LJ-13 and LJ-55. The results demonstrate that our method consistently outperforms FAB [1].
> > >
> > > ###### **Table 3. Comparison of wasserstein-2 distance of positions on LJ-13 and LJ-55 potentials.**
> > > | Method | LJ-13 | LJ-55  |
> > > | - | - | - |
> > > | FAB | 4.35 | 18.03 |
> > > | SGDS (ours) | $\mathbf{1.63}$ | $\mathbf{4.17}$ |
> > >
> > > You also noted Volokhova et al. [11], which is indeed relevant to our work. Their focus is on applying continuous GFN to conformation sampling by carefully designing the MDP, whereas our contribution lies in the general-purpose training algorithm of continuous GFN. These directions are orthogonal, and we will include a discussion in the manuscript.
> > >
> > > [11] Alexandra Volokhova et al., "Towards equilibrium molecular conformation generation with GFlowNets", Digital Discovery 2024

---

> > > > ### Comment · Reviewer_DVcd · 2025-08-08
> > > >
> > > > Thank you for the follow-up. I will keep my rating as an 'accept'.

---

### Official Review · Reviewer_Kvbn · 2025-07-10

**Clarity:** 3
**Significance:** 2
**Originality:** 3
**Rating:** 4
**Confidence:** 4

**Summary:**

This paper addresses the problem of sampling from an unnormalized energy with learned diffusion samplers. They combine an MCMC-based “searcher” with intrinsic novelty reward and a combination of offline and online diffusion sampler training using the trajectory balance loss. By combining these strategies, they are able to show improved results compared to previous diffusion sampler approaches (iDEM, PIS, TB) on several classical energies and a small peptide neural-network energy function.

**Questions:**

-Since the ground-truth is also generated by some MCMC variant, why wouldn’t we just do flow/score matching onto the MCMC samples and achieve better performance? How good do these proposal samples from MCMC need to be to effectively learn a diffusion sampler with the TB loss?

- What is the run-time compared to other algorithms? About how much time is spent on MCMC during each “searcher” phase? A run-time comparison would be helpful, especially for the case of Alanine dipeptide which uses the neural network energy function.


- How is the ELBO evaluated for diffusion samplers? It’s my understanding that for samplers like PIS, you use the path-weights via the Radon-Nikodyn derivative, which is not exactly the model-likelihood of the sample x_1.

**Ethical Concerns:**

["NO or VERY MINOR ethics concerns only"]

**Final Justification:**

The authors have provided additional experiments and clarified most of my concerns about the comparison to MCMC+flow-matching. The results are interesting and I like the idea of adding some MCMC strategy in the inner-loop. I do wish that the RND exploration loss provided a more principled understanding of addressing mode exploration / coverage, as it is somewhat heuristic from my understanding. That is why I lean towards weak accept rather than accept.

**Limitations:**

yes

**Quality:**

3

**Strengths And Weaknesses:**

Strengths:

- This paper combines MCMC sampling and offline diffusion sampler training in a novel way that yields good results on classical energy benchmarks which previous diffusion samplers have struggled with.

- The incorporation of an intrinsic reward in the energy for exploration is a novel heuristic for diffusion samplers (despite concerns below).

- This work presents many empirical evaluations of combining a number of interesting heuristics that seem to address mode collapse issues common with diffusion samplers.

Weaknesses:

- The abstract mentions addressing the limitation of “expensive energy functions”, where Manywell and Lennard Jones are very cheap to evaluate. It seems like the MCMC-based searcher might not scale well for expensive energies, and I suspect the majority of the run-time would be spent on this phase. It would be good to see neural network energy examples other than Alanine-dipeptide and their run-times (see [1] for results on larger peptides).

- I’m not very convinced by the heuristic “searcher” exploration reward as a contribution. The mean squared error to this random function at some point x will depend on things like the Lipschitz constant. It could be a good heuristic for novelty in the beginning of training, but won’t prevent that mode from being lost once it has been visited. In the ablation on Manywell 128, “w/o RND-searcher” performs only marginally worse than with RND.

[1] Tan et al 2025 “Scalable Equilibrium Sampling with Sequential Boltzmann Generators”

---

> ### Author Rebuttal · Authors · 2025-07-30
>
> Dear Reviewer Kvbn,
>
> Thank you for your thoughtful and encouraging review. We appreciate your comments on the novelty of the methodological incorporation and several evaluations of our work. Below, we address your points in detail.
>
> ### **W1: It seems like the MCMC-based searcher might not scale well for expensive energies. It would be good to see neural network energy examples other than Alanine Dipeptide and their run-times.**
>
> To resolve your concern, we provide additional results on Alanine Tripeptide and Tetrapeptide considered in Tan et al.[1].
>
> Table 1 shows that the searcher is not the bottleneck for scaling to larger systems. Table 2 shows that our method still outperforms MLE (or forward KL) training on samples produced by the searcher. All model architectures and training settings for the tripeptide and tetrapeptide follow those used for the dipeptide, with the only change being batch sizes of 8 for the tripeptide and 4 for the tetrapeptide (versus 16 for the dipeptide). Upon further inspection, we observed that our model successfully generates low‑energy conformations, and we plan to include 3D visualizations of these structures in the future manuscript.
>
> ###### **Table 1. Runtime (minutes) of neural network energy for a single round using NVIDIA GeForce RTX 3090.**
>
> | Task | Searcher | Learner | Total |
> | - | - | - | - |
> | Alanine Dipeptide | 128 | 1181 | 1309 |
> | Tripeptide | 139 | 2497 | 2636 |
> | Tetrapeptide | 157 | 3858 | 4015 |
>
> ###### **Table 2. Performance comparison with MLE on Alanine Tripeptide with the same energy calls (1M).**
>
> | Tripeptide | Gap $[\times 10^3]\downarrow$ | ELBO $[\times 10^3]\uparrow$ | EUBO $[\times 10^3]\downarrow$ |
> | - | - | - | - |
> | MLE | $6.117 \pm 0.019$ | $858.947 \pm 0.015$ | $865.064 \pm 0.002$ |
> | **Ours** | $\mathbf{5.600 \pm 0.030}$ | $\mathbf{859.455 \pm 0.031}$ | $\mathbf{865.055 \pm 0.005}$ |
>
> ###### **Table 3. Performance comparison with MLE on Alanine Tetrapeptide with the same energy calls (1M).**
> | Tetrapeptide | Gap $[\times 10^3]\downarrow$ | ELBO $[\times 10^3]\uparrow$ | EUBO $[\times 10^3]\downarrow$ |
> | - | - | - | - |
> | MLE | $29.072 \pm  0.560$ | $1,115.880 \pm  0.560$ | $1,144.953 \pm  0.002$ |
> | **Ours** | $\mathbf{26.600 \pm 0.214}$ | $\mathbf{1,118.342 \pm 0.213}$ | $\mathbf{1,144.942 \pm 0.013}$ |
>
> ---
>
> ### **W2-1: The Searcher exploration reward with RND won't prevent the previously visited mode from being lost once it has been visited.**
>
> It seems that you are concerned about the following case: when the RND method assigns too small exploration reward for a previously visited mode, the Learner loses this mode during training.
>
> Our algorithm avoids this issue for several reasons. First, in our training, the RND reward decays slowly even for previously visited modes, so that the Learner properly learns the mode before the RND reward becomes negligible. Next, we use a top-k priority buffer to keep the low energy states, hence the Learner does not lose the mode. Finally, even if the RND reward becomes negligible, the Searcher will have a relatively high probability of re-visiting the mode due to its low energy.
>
> ### **W2-2: In the ablation on Manywell 128, “w/o RND-searcher” performs only marginally worse than with RND.**
>
> The improvement is significant since it is in log scale. In the ablation on Manywell 128, the RND searcher improves ELBO by $6.4$ of ELBO, which actually represents $\exp(6.4) \approx 403.43$ times better performance in lower bounding the marginal probability of the Learner generating the true sample.
>
> ---
>
> ### **Q1-1: Since the ground-truth is also generated by some MCMC variant, why wouldn’t we just do flow/score matching onto the MCMC samples and achieve better performance?**
>
> Our algorithm outperforms flow/score matching on MCMC samples since our online learning collects informative samples in a sample-efficient manner with a good exploitation-exploration trade-off. That is, once the Learner is sufficiently trained, the Learner generates low-energy samples that are hard to collect from short MCMC trajectories (exploitation), while the Searcher is still necessary to collect samples from unseen modes (exploration).
>
> We already verified this in Figure 2-\(c\), where TB loss outperforms MCMC + MLE equivalent to score matching on the MCMC samples. Also, we show the debiasing effect of TB on Manywell distributions in Figure 6 in Appendix B.2.
>
> ### **Q1-2: How good do these proposal samples from MCMC need to be to effectively learn a diffusion sampler with the TB loss?**
>
> For Manywell 128 case, 300 parallel chains for Annealed Importance Sampling (AIS) with length 100 are enough to help the model learn well. For LJ-55, a single chain for Metropolis-Adjusted Langevin Algorithm (MALA) with length 10K is enough to learn well. For Alanine Dipeptide, four parallel chains of high-temperature molecular dynamics (MD) with length 110K are enough to capture diverse modes and train the learner well.
>
> ---
>
> ### **Q2: A run-time comparison would be helpful, especially for the case of Alanine Dipeptide, which uses the neural network energy function.**
>
> We present the table of runtime of the searcher and the learner in Alanine Dipeptide. Table 1 shows that the run-time of Searcher is marginal compared to that of Learner.
>
> ###### **Table 4. Runtime (minutes) of neural network energy using NVIDIA GeForce RTX 3090. Note that the first round searcher requires computing only neural network energy (51 minutes), but the second round searcher utilizes the RND network additionally (128 minutes).**
>
> | Alanine Dipeptide | Run-time |
> | - | - |
> | MLE | 1,583 |
> | **Ours** | 1,850 (searcher 179 + learner 1671) |
>
> ---
>
> ### **Q3: How is the ELBO evaluated for diffusion samplers? It’s my understanding that for samplers like PIS, you use the path-weights via the Radon-Nikodym derivative, which is not exactly the model-likelihood of the sample $x_1$.**
>
> We evaluate the ELBO by the following steps:
>
> * Compute $\log P_F(x_{t+\Delta t}|x_t ; \theta)$ along the forward path with the drift $u_\theta(x_t,t)$ and the noise schedule $\sigma$, and $\log P_B(x_t|x_{t+\Delta t})$ along the Brownian bridge for $t\in[0,1]$.
> * Average $\log\frac{R(x_1)\sum P_B(x_t|x_{t+\Delta t})}{\sum P_F(x_{t+\Delta t}|x_t ; \theta)}$ along the trajectories generated by the Learner.
>
> We follow prior works that use the ELBO as a performance metric [2-4] as a lower bound to the model likelihood of samples.
>
> ---
>
> [1] Tan et al., "Scalable Equilibrium Sampling with Sequential Boltzmann Generators", ICML 2025.
>
> [2] Sendera et al., "Improved off-policy training of diffusion samplers", NeurIPS 2024.
>
> [3] Kim et al., "Adaptive teachers for amortized samplers", ICLR 2025.
>
> [4] Vargas et al., "Transport meets variational inference: Controlled Monte Carlo diffusions", ICLR 2024.
>
> ---
> We hope our responses have addressed your questions and concerns. Please let us know if there is anything we may have missed or if any points require further clarification.

---

> ### Comment · Reviewer_Kvbn · 2025-08-05
>
> I want to thank the authors for their clarifications and their efforts for addressing my concerns.
>
> One last question regarding the result about MCMC+flow-matching, so I can understand it correctly. Shouldn't just running MCMC longer completely solve LJ55 since that is what generated the ground truth? I would think that fitting a generative model to MCMC data run on a longer chain would eventually do better than your proposed method. So your result is showing that with some fixed energy call budget, your method performs better?
>
> That being said, I will raise my score accordingly.

---

> > ### Author Response · Authors · 2025-08-05
> >
> > We thank the reviewer for the insightful follow-up question. You’re right about the long-run behavior: if we run MCMC long enough on LJ55, it will recover the target distribution, and training a flow-matching model to imitate on those samples will also work well. The practical issue is **sample efficiency**. Since many methods converge with unlimited budget, it only makes sense to compare them **under a fixed number of energy evaluations**. In this condition SGDS outperforms MCMC + imitation learning methods.
> >
> > The reason is that imitation only learns from what’s in the offline MCMC set, while **off-policy RL (ours) learns from the energy to reweight, explore, and correct regions** that the dataset under-covers.
> >
> > To further illustrate this point, in the ablation study in Fig. 2(c), we tested your hypothesis on Manywell-128 by doubling the searcher’s trajectory length and increasing the number of parallel chains by 1.5×, followed by amortization via imitation learning (MLE training) of the diffusion model. Even under these more generous conditions, SGDS still consistently outperforms longer MCMC + MLE, reinforcing our claim about the practical efficiency and effectiveness of SGDS.
> >
> > Note that the Manywell-128 benchmark does not require high model expressivity, so the performance of MLE-based methods primarily reflects the quality of the MCMC samples rather than the expressive power of the generative model (e.g., diffusion vs. flow matching).

---

> > > ### Comment · Reviewer_Kvbn · 2025-08-08
> > >
> > > Thank you for the clarification. I think most of my concerns are addressed. It would be great to see more ablation performed on the RND weighting in the final paper to support the paper's algorithmic novelty. I also think the comparison to Adjoint Sampling found in the other responses is interesting and should be included.
> > >
> > > I remain positive about this work, but would like to have more discussions with the reviewers before raising my score further.

---

> > > > ### Author Response · Authors · 2025-08-09
> > > >
> > > > We are glad that most of your concerns have been addressed.
> > > > From your comment, it seems that you have seen some of the RND weight ablation results provided in our responses to other reviewers. To  take your advice into account for a more detailed analysis, we conducted additional experiments with three extra RND weight values.
> > > >
> > > >
> > > > ###### **Table 5. ELBO-EUBO gap at 20% of the second round for Manywell-128.**
> > > > | RND weight | $1$ | $10$ | $50$ |$100$ | $150$ | $200$ | $1000$ |
> > > > | - | - | - | - | - | - | - | - |
> > > > | Manywell-128 | 549.54 | 542.60 | 543.16 | **538.79** | 540.77 | 544.15 | 570.49 |
> > > >
> > > >
> > > > In addition to the values already compared in the rebuttal (10, 100, 1000), we now include results for nearby values (50, 150, 200) around the final choice (100). Due to the limited time available, we ran these additional experiments only on the Manywell-128, but we will include more detailed ablations over all tasks in the future manuscript.
> > > >
> > > > We also agree that the comparison to Adjoint Sampling is valuable, and we will ensure that it is included in the final version of the paper.

---

### Note · Authors · 2025-08-13

Dear Area Chair and Reviewers,

We sincerely thank you for your time and effort in reviewing our manuscript. We are encouraged by the reviewers’ positive assessments and constructive engagement during the rebuttal phase. To assist in your final evaluation, we summarize below the key discussion points and improvements made during the rebuttal.

* To resolve the concerns of Reviewer Kvbn, we demonstrated the practical significance and robustness of our method on larger systems by conducting additional experiments with tripeptides and tetrapeptides.

* Following the suggestions by Reviewer Dvcd and HM7Y, we evaluated our method against Adjoint Sampling and Flow Annealed Importance Sampling Bootstrap (FAB), showing that our approach consistently outperforms these baselines.

* In response to Reviewer Dvcd and HM7Y, we showed that RND weight does not require much hyperparameter tuning.

All rebuttal clarifications and new experimental results will be incorporated into the final manuscript. We believe the rebuttal process has substantially strengthened our work.

Thank you once again for your constructive feedback and guidance.

Sincerely,
Authors.

---

### Decision · Program_Chairs · 2025-09-17

**Decision:**

Accept (poster)

**Comment:**

The paper introduces a novel diffusion sampler. The algorithm is a combination of an MCMC searcher with intrinsic novelty reward and offline and online diffusion sampler training using the trajectory balance loss.
The paper is well written and it combines ideas from sample-efficient RL with MCMC. The method is showcased in several benchmarks in the paper and during the rebuttal phase, as response to reviewers' questions.
Some of the concerns raised are about the conceptual novelty, runtime, and the practicality on real world tasks. The paper however focuses on tasks that are standard in the neural sampler literature, and does a through comparison with state of the art methods, showing consistent improvements.